# Rethinking Addressing in Language Models via Contextualized Equivariant Positional Encoding

## Abstract

Transformers rely on both content-based and position-based addressing mechanisms to make predictions, but existing positional encoding techniques often diminish the effectiveness of position-based addressing. Many current methods enforce rigid patterns in attention maps, limiting the ability to model long-range dependencies and adapt to diverse tasks. Additionally, most positional encodings are learned as general biases, lacking the specialization required for different instances within a dataset. To address this, we propose TAPE: conTextualized equivariAnt Position Embedding, a novel framework that enhances positional embeddings by incorporating sequence content across layers. TAPE introduces dynamic, context-aware positional encodings, overcoming the constraints of traditional fixed patterns. By enforcing permutation and orthogonal equivariance, TAPE ensures the stability of positional encodings during updates, improving robustness and adaptability. Our method can be easily integrated into pre-trained transformers, offering parameter-efficient fine-tuning with minimal overhead. Extensive experiments shows that TAPE achieves superior performance in language modeling, arithmetic reasoning, and long-context retrieval tasks compared to existing positional embedding techniques.

## 1 Introduction

Attention mechanisms are a core component of many modern deep learning architectures, enabling models to selectively focus on relevant information within a given context. Transformer models (Vaswani et al., 2017) and their numerous variants (Carion et al., 2020; Dosovitskiy et al., 2021; Zhao et al., 2021), which are fundamentally driven by attention, have revolutionized tasks involving sequential and spatial data, such as text (Kitaev et al., 2020), image (Dosovitskiy et al., 2021), and point cloud (Zhao et al., 2021). More recently, large transformer models have become dominant in natural language understanding, language generation, and complex reasoning (Brown et al., 2020).

Delving into underlying computational paradigm of attention, the prediction made for each token is expressed as a weighted aggregation over the representations of other tokens. Due to the nature of the softmax function, attention often generates a sparse mask, extracting a limited subset of tokens for interaction. Through this interpretation, attention can be understood as an *addressing* mechanism that searches the context, locating and retrieving token representations deemed most relevant or important. Since the attention score is computed upon token features and positions (see Section 2), transformers' addressing ability is based on two fundamental mechanisms: *content-based* addressing and *position-based* addressing. Content-based addressing is accomplished by recognizing relevant tokens through feature similarity, while position-based addressing is facilitated by positional encoding techniques, which are designed to ideally enable random access along the sequence via indexing. It is more important to let them cooperate to tackle more complex tasks, such as in-context retrieval (Hinton & Anderson, 2014; Ba et al., 2016), arithmetic (Lee et al., 2023; McLeish et al., 2024b), counting (Golovneva et al., 2024), logical computation (Liu et al., 2024), and reasoning (Wei et al., 2022; Rajani et al., 2019; Dziri et al., 2024). However, we contend that the role of position-based addressing is diminished and limited in most transformer architectures (Ebrahimi et al., 2024).

It has not escaped our notice that most existing positional encodings weakens the position-based addressing capability. Recent works (Press et al., 2021b; Su et al., 2024; Chi et al., 2022b; Sun et al., 2022) impose a fixed and somewhat artisanal pattern on attention maps, typically adopting a decaying pattern in relation to relative distances, thereby enforcing a locality bias. This rigidity limits the ability of positional encodings to model long-range dependencies and makes it challenging to attend to distant query-key pairs. Although some positional encodings are parameterized trainable parameters (Vaswani et al., 2017; Shaw et al., 2018; Chi et al., 2022a; Li et al., 2023), the hypothesis space is often excessively constrained. Perhaps more crucially, most existing positional encodings are designed and learned as a general bias across the entire dataset, lacking specialization and adaptability to specific instances informed by the context. The interplay between context and positional embeddings has proven essential in LLMs for various compositional tasks such as algorithmic (McLeish et al., 2024a), language modeling and coding tasks (Golovneva et al., 2024). Recent studies indicate that token indices can be reconstructed through causal attention, suggesting the elimination of positional encoding (Haviv et al., 2022; Wang et al., 2024b; Kazemnejad et al., 2024). However, their arguments require a specific configuration of transformer weights, which may not be achievable.

To unleash the power of position-based addressing, we endeavor to design a more universal and generic position encoding for language transformers. We introduce Contextualized Equivariant Positional Encoding (TAPE), a novel framework designed to contextualize positional embeddings by incorporating sequence content. Our TAPE continually progresses information flow between positional embeddings and token features via specialized attention and MLP layers. To ensure the stability of positional encodings during model updates, we enforce permutation and orthogonal group equivariance properties on attention and MLP layers. This enforcement guarantees robustness to input permutations and translations on sequences, and maintains relative relationships between encoded positions, further strengthening the model's capacity to generalize across diverse domains.

Technically, we extend conventional vectorized positional embeddings into a multi-dimensional tensor, which enriches interactions between positional embeddings and token features. In the attention mechanism, TAPE incorporates the pairwise inner product between positional encodings, allowing the attention values to be computed based on not only token similarities but also positional relationships. The resulting attention map carrying token correlations is further used to inform positional features through a linear combination. In addition to the attention mechanism, we also customize an MLP layer that directly mixes token features with positional encodings, while preserving orthogonal equivariance.

We demonstrate the superior performance of TAPE on arithmetic reasoning tasks (McLeish et al., 2024a), which require LLMs to effectively locate/address and retrieve specific tokens, as well as on representative natural language tasks, including SCROLLS (Shaham et al., 2022) and passkey retrieval (Mohtashami & Jaggi, 2023), to validate the generalizability of the framework.

Our contributions are summarized as follows:

- We introduce TAPE, a novel framework to contextualize positional embeddings with sequence content across layers to enhance the position-addressing ability of transformers. We further enforce TAPE with permutation and orthogonal equivariance to guarantee the stability of positional encodings during the update.

- We propose practical implementations for our TAPE, which extends conventional positional embeddings into multi-dimensional and facilitates attention and MLP in transformers with two levels of equivariance. We also show that TAPE can be used as a drop-in component into extant pre-trained models for parameter-efficient fine-tuning.

- We conduct extensive experiments, showcasing TAPE is superior in both training from scratch and parameter-efficient fine-tuning scenarios for language modeling as well as downstream tasks such as arithmetic reasoning and long-context retrieval. We show that TAPE achieves state-of-the-art performance in language modeling tasks, surpassing baselines in perplexity reduction for long sequences. We also report the state-of-the-art performance of TAPE in long-context tasks like passkey retrieval tasks with LLM fine-tuning and addition tasks with arithmetic learning.

## 2 PRELIMINARIES

In this work, we aim to design expressive and generalizable positional embeddings for transformers to address complex language tasks. Let $\boldsymbol{X} = [\boldsymbol{x}_1 \cdots \boldsymbol{x}_N]^\top \in \mathbb{R}^{N \times C}$ represent the input sequence of tokens, where $N$ is the context length and $C$ is the feature dimension. Transformers learn token representations using the attention mechanism (Vaswani et al., 2017), which propagates information across tokens by computing pairwise correlations. Since pure attention is inherently permutation-equivariant, language models integrate positional information into the attention computation to differentiate tokens based on their positions.

### 2.1 HIGH-DIMENSIONAL FEATURES AS POSITIONAL ENCODING

One common approach is to leverage high-dimensional features to represent positions. Denote positional encoding as $\boldsymbol{E} = [\boldsymbol{e}_1 \cdots \boldsymbol{e}_N] \in \mathbb{R}^{N \times D}$, where $D$ represents the embedding dimension. When computing the attention value, the pre-softmax attention value can be in general formulated as [1]:

$$\alpha_{i,j} = q(\boldsymbol{x}_i, \boldsymbol{e}_i)^\top k(\boldsymbol{x}_j, \boldsymbol{e}_j), \tag{1}$$

where $q(\cdot, \cdot)$ and $k(\cdot, \cdot)$ are generalized query and key transformations that incorporate positional features. In the original transformer paper (Vaswani et al., 2017), $\boldsymbol{E}$ assigns each absolute position an either learnable or fixed sinusoidal embedding. The query and key transformations directly add the positional information into token features at the first layer: $q(\boldsymbol{x}, \boldsymbol{e}_i) = \boldsymbol{W}_Q(\boldsymbol{x} + \boldsymbol{e}_i)$ and $k(\boldsymbol{x}, \boldsymbol{e}_j) = \boldsymbol{W}_K(\boldsymbol{x} + \boldsymbol{e}_j)$ for some query and key matrices $\boldsymbol{W}_Q, \boldsymbol{W}_K \in \mathbb{R}^{F \times C}$. Shaw et al. (2018) introduces learnable embeddings for relative distances, which are applied to the key vector during attention computation. More recently, Rotary Position Encoding (RoPE) (Su et al., 2024) has gained widespread adoption in modern LLMs (Touvron et al., 2023a;b; Biderman et al., 2023; Chowdhery et al., 2023; Jiang et al., 2023). RoPE encodes absolute positions using block-wise rotation matrices, while implicitly capturing relative distances during dot-product attention. RoPE defines the positional embeddings and the transformation $q(\cdot, \cdot)$ as shown below, with $k(\cdot)$ adhering to a similar formulation:

$$q(\boldsymbol{x}, \boldsymbol{e}_i) = [\boldsymbol{q}_1 \odot \boldsymbol{e}_{cos,i} - \boldsymbol{q}_2 \odot \boldsymbol{e}_{sin,i} \quad \boldsymbol{q}_1 \odot \boldsymbol{e}_{sin,i} + \boldsymbol{q}_2 \odot \boldsymbol{e}_{cos,i}]^\top, \quad \boldsymbol{q} = \boldsymbol{W}_Q \boldsymbol{x}, \tag{2}$$

where $\odot$ denotes element-wise multiplication. RoPE equally divides query feature $\boldsymbol{q} = [\boldsymbol{q}_1 \quad \boldsymbol{q}_2]^\top$ into the real and imaginary components, and represents $\boldsymbol{e}_i = [\boldsymbol{e}_{cos,i} \quad \boldsymbol{e}_{sin,i}]^\top, i \in [N]$ as cosine and sine series: $\boldsymbol{e}_{\omega,i} = [\omega(\theta_1 i) \quad \cdots \quad \omega(\theta_{D/2} i)]^\top$ where $\omega \in \{\cos, \sin\}$, and $\theta_d = -10000^{2d/D}, d \in [D/2]$. Subsequent works explore methods to extend the context length for RoPE-based LLMs through the adoption of damped trigonometric series (Sun et al., 2022), positional interpolation (Chen et al., 2023a) and adjustments to coefficients $\{\theta_d\}$ (r/LocalLLaMA, 2023; Peng et al., 2023; Liu et al., 2023).

### 2.2 ATTENTION BIAS AS POSITIONAL ENCODING

An alternative method for encoding positional information involves applying a bias to the attention map, conditioned on the relative distances between tokens during the attention computation. The pre-softmax attention value with bias can be formulated as:

$$\alpha_{i,j} = (\boldsymbol{W}_Q \boldsymbol{x}_i)^\top (\boldsymbol{W}_K \boldsymbol{x}_j) + b(i,j), \tag{3}$$

where $b(i,j) : \mathbb{N} \times \mathbb{N} \to \mathbb{R}$ is a bias regarding the token indices $i$ and $j$. Many existing positional encoding methods can be interpreted as various instantializations of $b(i,j)$. We follow Li et al. (2023) to summarize a few examples below:

- In T5 (Raffel et al., 2020), $b(i,j) = r_{\min\{i-j, L_{max}\}}$, where $L_{max}$ denotes the maximal relative distance considered, and $\{r_i \in \mathbb{R} : i \in [0, L_{max}]\}$ are learnable scalars.
- Alibi (Press et al., 2021b) simplifies the bias term to $b(i,j) = -r|i - j|$, where $r > 0$ is a hyperparameter that acts as the slope, imposing a linear decay pattern based on the relative distance.

---

[1] For simplicity, we ignore the denominator $\sqrt{F}$ by default.

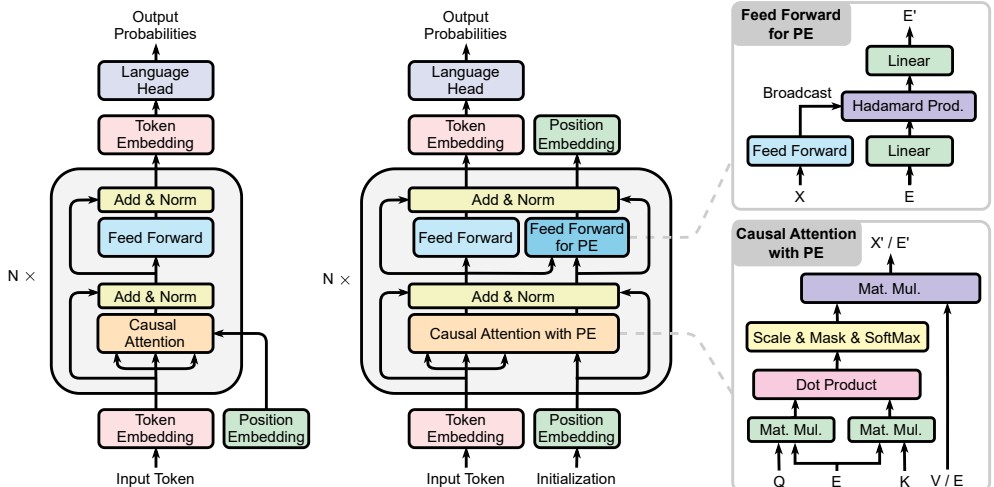

(a) Traditional position embedding.    (b) TAPE with enhanced causal attention and feed forward layers.

Figure 1: Overview of our proposed TAPE in standard decoder-only Transformer architecture.

- Kerple (Chi et al., 2022a) enforces a logarithmic or power decay rate: $b(i,j) = -r_1 \log(1+ r_2|i-j|)$ and $b(i,j) = -r_1|i-j|^{r_2}$ respectively, where $r_1, r_2 > 0$ are hyperparameters.
- FIRE (Li et al., 2023) learns a neural network with parameters $\boldsymbol{\theta}$ to model the bias: $b(i,j) = f_{\boldsymbol{\theta}}(\psi(i-j)/\psi(\max\{i, L\}))$, where $\psi(x) = \log(cx+1)$, and $L > 0$ is a hyperparameter.

## 3 OUR APPROACH

### 3.1 MOTIVATIONS AND DESIGN PRINCIPLES FOR POSITION ENCODING

In the paper, we interpret the attention mechanism as an addressing system, where row-wise attention logits can be viewed as an indicator vector locating important tokens in the context to inform predictions for the current token. The underlying addressing mechanisms include both content-based addressing, which locates tokens via feature similarity, and position-based addressing, which leverages positional encodings to extract location-based information. Content-based addressing is often prioritized in language modeling – which is evidenced by a series of simplifications on positional encoding in the literature (Press et al., 2021b; Haviv et al., 2022; Wang et al., 2024b; Kazemnejad et al., 2024) – due to the fact that natural language semantics primarily depend on the meaning of constituent words rather than their arrangement order (Sinha et al., 2021). However, position-based addressing can sometimes be crucial for many advanced tasks. Ebrahimi et al. (2024) demonstrates that in arithmetic tasks (Lee et al., 2023), a token's position is as important as its value. Specifically, an ideal attention map for performing addition needs to exclusively rely on token indices.

Moreover, we observe that the interaction between token features and positional embeddings is lacking in current transformer models. Golovneva et al. (2024) demonstrate that incorporating the interplay between context and positional information allows for more flexible addressing, leading to improvements in complex compositional tasks such as algorithm execution and logical reasoning (Liu et al., 2024).

Based on above arguments, we aim to establish a more expressive positional encoding scheme, which can be effectively informed by the context to facilitate position-based addressing in LLMs. The main idea is to customize attention and MLP modules in transformers such that they can update positional embeddings at each layer with sequence content, and use the updated embeddings as the positional encoding for the next layer.

Let a tuple $(\boldsymbol{X}, \boldsymbol{E})$ represent a language sequence, where $\boldsymbol{X} \in \mathbb{R}^{N \times C}$ are the token features, $\boldsymbol{E} \in \mathbb{R}^{N \times D}$ are the positional embeddings. We define a transformer block consisting of two separate embedding layers: token mixing layer and position contextualizing layer. The token mix-

ing layer is formulated as a function $f : \mathbb{R}^{N \times C} \times \mathbb{R}^{N \times D} \to \mathbb{R}^{N \times C}$, which combines token features and positional embeddings to represent each token. The position contextualizing layer $g : \mathbb{R}^{N \times C} \times \mathbb{R}^{N \times D} \to \mathbb{R}^{N \times D}$ encodes the context information into the positional embeddings. We establish two fundamental criteria for the design of both functions. Conceptually, by representing each token as a tuple comprising its token and positional embedding, the entire sequence can be viewed as an unordered set. This implies that permuting these tuples arbitrarily will not alter the outputs of $f$ and $g$, aside from a corresponding change in order (Zaheer et al., 2017; Lee et al., 2019). We note that this is naturally satisfied by attention. Furthermore, we aim for the positional embeddings to effectively model relative distances, necessitating that $f$ remains invariant to translations in the token positions (Sun et al., 2022). As will be demonstrated later, this invariance can be achieved by structuring $f$ to depend on the positional embedding in a manner invariant to orthogonal transformations. In the context of updating positional features via $g$, we seek to maintain their internal geometric structures, which we accomplish by ensuring that $g$ undergoes the same transformation when the positional embedding inputs are subjected to an orthogonal matrix (Villar et al., 2021). Enforcing orthogonal invariance for $f$ and $g$ is critical to achieve numerical stability (Wang et al., 2022; Huang et al., 2023), enabling the representation of a sequence to remain consistent under positional translation (Sun et al., 2022).

Formally, let us denote $\Pi(N)$ as a permutation group, and $O(D)$ as an orthogonal group. The two aforementioned criteria require $f$ and $g$ to satisfy the following two equations:

$$f(\boldsymbol{PX}, \boldsymbol{PER}) = \boldsymbol{P}f(\boldsymbol{X}, \boldsymbol{E}), \quad \forall \boldsymbol{P} \in \Pi(N), \boldsymbol{R} \in O(D), \tag{4}$$

$$g(\boldsymbol{PX}, \boldsymbol{PER}) = \boldsymbol{P}g(\boldsymbol{X}, \boldsymbol{E})\boldsymbol{R}, \quad \forall \boldsymbol{P} \in \Pi(N), \boldsymbol{R} \in O(D). \tag{5}$$

### 3.2 TAPE: Contextualized Positional Encoding with Equivariance

In this section, we instantiate design principles discussed in Sec. 3.1 as a practical neural architecture. We note that although there are lots of ways to achieve conditions in Eq. 4 and 5 (Dym & Maron, 2020; Bogatskiy et al., 2020; Yarotsky, 2022), the proposed method focuses on enhancing existing components used in standard transformers with consideration of computational efficiency. We term our proposed approach of informing positional encoding with context through enforcing equivariance as ConTexturalized EquivAriant Positional Encoding (TAPE).

**Tensorial Positional Encoding.** Our first enhancement involves extending positional encodings to a multi-dimensional format, facilitating diverse interactions with token features. Traditionally, positional encoding is represented as a vector for each token. In contrast, we propose dividing the channel dimension of each token into $M$ segments and assigning a matrix-form positional embedding to each block. Formally, if $C = MB$, the sequence of token features can be reshaped to $\boldsymbol{X} \in \mathbb{R}^{N \times M \times B}$. Each block is then allocated an $L \times D$ matrix as its positional encoding. All positional embeddings can be collectively organized as a tensor $\boldsymbol{E} \in \mathbb{R}^{N \times M \times L \times D}$. This design intuitively interprets each token as comprising $M$ smaller information units, each equipped with $L$ sets of $D$-dimensional coordinates. As a result, the attachment between positional embeddings and token features becomes more flexible and diversified. Our tensorial positional encoding draws inspiration from, yet also generalizes, the positional encoding representations presented in Deng et al. (2021) and Wang et al. (2024a). We will enforce permutation-equivariance over the first dimension (of size $N$), while ensure $O(D)$-invariance/equivariance over the last dimension of $\boldsymbol{E}$ (with size $D$).

**Model Structure and Initialization.** We adhere to the conventional architecture of the standard transformer, wherein each layer comprises an attention module for token mixing and a Multi-Layer Perceptron (MLP) for channel mixing. However, the whole model takes both token and positional embeddings as inputs (akin to the original transformer (Vaswani et al., 2017)). In the meanwhile, both the attention and MLP components are tailored to update positional embeddings at each layer. The initial positional features may encompass a variety of representations, including but not limited to learnable features (Vaswani et al., 2017), sinusoidal series (Vaswani et al., 2017; Su et al., 2024; Sun et al., 2022), or random Fourier features (Rahimi & Recht, 2007; Yu et al., 2016).

**Token Mixing.** In each transformer block, $f$ updates token features through attention and an MLP following the principles of permutation-equivariance and $O(D)$-invariance. We define pre-softmax

attention value between the $i$-th and $j$-th tokens as:

$$\alpha_{i,j} = \sum_{m=1}^{M} \alpha_{i,j,m}, \quad \alpha_{i,j,m} = (\boldsymbol{W}_{Q,m} \boldsymbol{x}_{j,m})^{\top} \phi(\boldsymbol{e}_{j,m}^{\top} \boldsymbol{e}_{i,m})(\boldsymbol{W}_{K,m} \boldsymbol{x}_{i,m}), \quad (6)$$

where $\phi(\cdot) : \mathbb{R}^{L \times L} \to \mathbb{R}^{B \times B}$ can be any function. Permutation-equivariance is inherently preserved in pairwise attention, regardless of the method used to derive attention values. $O(D)$-invariance is achieved by computing the inner product of positional embeddings (Villar et al., 2021). We note that $O(D)$-invariance stems from the separation of the inner product calculations between features and positional embeddings, in contrast to Vaswani et al. (2017). In practice, we can let $L = B$ and $\phi$ be an identity mapping, which simplifies Eq. 6 to a hardware-efficient tensor multiplication. After applying attention, a standard MLP is employed to further transform the features for each token without using positional encoding.

**Position Contextualization.** The primary contribution of this work is the introduction of a method to condition positional embeddings on sequence content. We employ an $O(D)$-equivariant function $g$ to ensure the stability of this update. A key insight is that linearly combining positional coordinates preserves $O(D)$-equivariance, provided the weights are invariant to the orthogonal group (Villar et al., 2021). This observation leads us to leverage attention maps, which capture content-based token relationships, to integrate positional embeddings. Henceforth, the attention layer can update positional embedding via:

$$\widetilde{\boldsymbol{e}}_{j,m} = \sum_{i=1}^{N} \frac{\exp(\alpha_{i,j,m})}{\sum_{i=1}^{N} \exp(\alpha_{i,j,m})} \boldsymbol{e}_{i,m}, \quad \forall j \in [N], m \in [M], \quad (7)$$

where $\tilde{\boldsymbol{e}}_{j,m}$ denotes an intermediate output of the attention layer. In practice, we share the attention map between Eq. 6 and 7. We can re-use $\alpha_{i,j,m}$ computed in Eq. 6 because attention weights computed for token mixing already achieves $O(D)$-invariance. We further propose an MLP-like layer to directly transform matrix-form positional embeddings with token features integrated. Specifically, each positional embedding is updated as:

$$\widehat{\boldsymbol{e}}_{j,m} = \boldsymbol{W}_2 \operatorname{diag}(\psi(\widetilde{\boldsymbol{x}}_{j,m})) \boldsymbol{W}_1 \widetilde{\boldsymbol{e}}_{j,m}, \quad \forall j \in [N], m \in [M], \quad (8)$$

where we denote $\widetilde{\boldsymbol{x}}_{j,m}$ as the output of attention used for token mixing, $\widehat{\boldsymbol{e}}_{j,m}$ as the final output positional encoding of the transformer block, $\psi : \mathbb{R}^B \to \mathbb{R}^{B'}$ can be arbitrary mapping chosen as an MLP in practice, $\operatorname{diag}(\cdot)$ constructs a diagonal matrix where the input vector is placed along the diagonal, with all off-diagonal elements set to zero, $\boldsymbol{W}_1 \in \mathbb{R}^{B' \times L}, \boldsymbol{W}_2 \in \mathbb{R}^{L \times B'}$ are trainable weight matrices, and $B'$ denotes the dimension of some intermediate hidden space. By applying these transformations to the left of the positional embedding, the process maintains $O(D)$-equivariance. Non-linear activations are applied through $\psi$ as they cannot directly act on positional embeddings. Here, we emphasize the importance of tensorial parameterization for positional encoding, as it introduces an additional dimension, enabling more complex transformations while preserving equivariance. Addtionally, we also introduce residual connections for positional embeddings while ignoring normalization layers upon them.

**Proposition 1.** *The proposed model including attention in Eq. 6 with normal MLP and attention in Eq. 7 with MLP defined in Eq. 8 satisfies Eq. 4 and Eq. 5.*

### 3.3 PARAMETER-EFFICIENT FINE-TUNING WITH TAPE

In this section, we demonstrate that our TAPE can be seamlessly integrated into pre-trained models, enabling parameter-efficient fine-tuning to enhance position-based addressing in existing architectures. Notably, the widely adopted RoPE (Su et al., 2024) can be considered a special case of TAPE. This can be seen by letting $L = D = 2$ and $\boldsymbol{e}_{i,m} = \begin{bmatrix} \cos(\theta_m i) & -\sin(\theta_m i) \\ \sin(\theta_m i) & \cos(\theta_m i) \end{bmatrix}$. With this configuration, Eq. 6 becomes equivalent to Eq. 2. As a result, RoPE can serve as the initialization for TAPE, while the model is further enhanced by incorporating the contextualization component specified in Eq. 7 and 8. To ensure the augmented model is identical to the original at the initialization, we set the initialization of $\boldsymbol{W}_2$ in Eq. 8 to all zeros following Hu et al. (2021). All updates to the positional encoding inside the block will then be reset via a residual connection.

## 4 EXPERIMENTS

In this section, we first validate our method on arithmetic tasks, which explicitly rely on absolute positions for prediction (Sec. 4.1). We also show our effectiveness in natural languages, in both pre-training (Sec. 4.2) and fine-tuning case (Sec. 3.3).

### 4.1 ARITHMETIC LEARNING

As demonstrated by prior research (Lee et al., 2023; Zhou et al., 2024), even large transformer models struggle with arithmetic tasks. Recent studies suggest that this limitation may stem from their constrained position-addressing capabilities (Ebrahimi et al., 2024). In particular, arithmetic tasks treat every digit as equally important to the equation, regardless of its distance from the output. In contrast, traditional positional embeddings for language tasks often assume a distance-decay effect, where words farther apart have less significance in the output. Positional contextualization potentially addresses this by dynamically reweighting positional importance based on the task context. To evaluate the ability of LLMs of performing arithmetic tasks with our position embedding, we use the Addition Bucket 40 dataset (McLeish et al., 2024a) which contains 20 million samples with $i \times i$ ( $i < 40$) operand lengths. We train transformers from scratch using the arthimetic data, and during evaluation, we sample 100 samples for each pair of operand lengths. Following the existing attempt (McLeish et al., 2024a), the operands in the training set are not necessary to have the same length, but the maximum length of two operands are the same. We then report model accuracy for each $(i, j)$ length pair. Note that accuracy is measured strictly, counting only exact matches of all output digits as correct. The transformers are standard decoder-only architecture with the number of layers 16, the hidden dimension 1024, intermediate dimension 2048 and the number of attention heads 16. The total number of model parameters is approximately 120M. We compare our method with four baselines, including RoPE (Kitaev et al., 2020), RandPE (Ruoss et al., 2023) NoPE (Kazemnejad et al., 2024), and FIRE (Li et al., 2023).

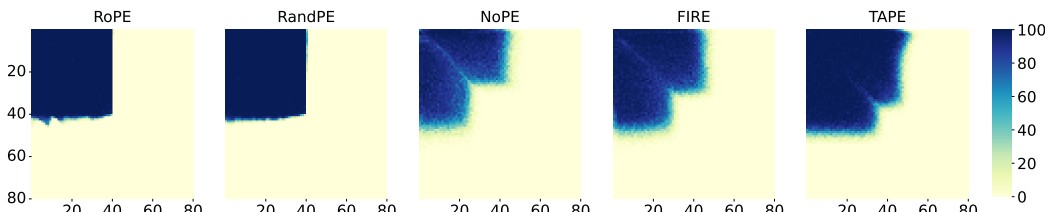

Figure 2: Accuracy on addition task between different methods on $2\times$ context length. Models are trained on sequence with length up to 40 while test on sequence with length up to 80. The average accuracy across the heatmap is 26.32%, 26.56%, 22.45%, 26.98% and 32.82% respectively for RoPE, RandPE, NoPE, FIRE and TAPE.

The heatmaps further demonstrate TAPE's superior generalization to longer sequences, as indicated by the concentrated dark-colored regions representing higher accuracy across a wider range of operand lengths. TAPE outperforms other methods with the highest average accuracy of 32.82%. Compared to FIRE, which achieves 26.98% and previously held the strongest length generalization in arithmetic tasks (McLeish et al., 2024a; Zhou et al., 2024), TAPE shows a remarkable 21.6% relative improvement. This shows TAPE's effectiveness in maintaining accuracy as sequence lengths increase, making it particularly suitable for long-range dependency tasks.

### 4.2 PRE-TRAINING FROM SCRATCH

Pre-training a language model on a corpus followed by fine-tuning on downstream tasks is the standard methodology for evaluating the performance of positional embeddings in prior studies (Li et al., 2023; He et al., 2024). Similarly, we first pre-train transformers with 1024 context window from scratch, using C4 dataset (Raffel et al., 2020), and then fine-tune those models in long-context benchmark SCROLLS (Shaham et al., 2022). We report three evaluation metrics for seven different tasks: unigram overlap (F1) for Qasper and NarrativeQA, and exact match (EM) for QuALITY (QAS) and ContractNLI (CNLI), and Rgm score (the geometric mean of ROUGE-1,2,L) for the three summarization tasks: GovReport (GovR), QMSum (QMS), and SummScreenFD (SumS).

Table 1: Performance comparison on seven datasets from SCROLLS benchmark.

| | QAS | CNLI | NQA | QuAL | QMS | SumS | GovR |
|---|---|---|---|---|---|---|---|
| Metric (%) | F1 | EM | F1 | EM | Rgm | Rgm | Rgm |
| Median length | 5472 | 2148 | 57829 | 7171 | 14197 | 9046 | 8841 |
| RoPE (Kitaev et al., 2020) | 8.39 | 65.00 | 1.77 | 0.04 | 6.34 | 5.63 | 9.71 |
| ALiBi (Press et al., 2021a) | 8.25 | 69.62 | 4.11 | 0.0 | 9.92 | 9.78 | 18.81 |
| RandPE (Ruoss et al., 2023) | 13.44 | 62.01 | 4.63 | 0.38 | 8.43 | 8.31 | 8.93 |
| FIRE (Li et al., 2023) | 3.41 | 71.26 | 0.48 | 1.25 | 8.78 | 7.42 | - |
| xPos (Sun et al., 2022) | 9.02 | 71.75 | 4.83 | 0.24 | 10.73 | 9.38 | 16.38 |
| TAPE (ours) | 11.52 | 72.80 | 6.79 | 11.60 | 12.42 | 10.34 | 15.18 |

We choose the standard decoder-only Transformer as the base model with the number of layers 12, the hidden dimension 768, intermediate dimension 3072, and the number of attention heads 12. The total number of model parameters is approximately 155M. We compare our methods with RoPE (Kitaev et al., 2020), ALiBi (Press et al., 2021a), RandPE (Ruoss et al., 2023), FIRE (Li et al., 2023) and xPos (Sun et al., 2022), and report the results in Table 1.

Our method consistently outperforms all baselines, with significant improvements especially in cases with longer context lengths, such as in QuAL and NQA. While FIRE achieves competitive results in CNLI and QuAL, its performance degrades in QAS and NQA. We speculate that this could be due to the optimization challenges of FIRE, as we observed its converged weights to be numerically near thresholds and sometimes slower to converge under our training recipe detailed in Appendix A.

## 4.3 Context Window Extension by Parameter-Efficient Tuning

We extend the context window of the pre-trained Llama2 7B model (GenAI, 2023) from 4096 to 8192, using the Redpajama (Computer, 2023). For validation, we then compare the perplexity on sequence of length 8192, on the cleaned ArXiv Math proof-pile dataset (Azerbayev et al., 2022; Chen et al., 2023a) and the book corpus dataset PG19 (Rae et al., 2019). To further evaluate the models' performance of long context understanding, we report the accuracy of fine-tuned models on passkey retrieval task which has been adopted by many literature (Chen et al., 2023b;a; Tworkowski et al., 2024). We choose a popular open-sourced large language model Llama2 7B (Touvron et al., 2023b) as the base model and extend it to the 8192 context length. Three baselines are selected to compare to our TAPE method: vanilla LoRA (Hu et al., 2022), LongLoRA (Chen et al., 2023b), Theta Scaling (Liu et al., 2023).

Table 2: Evaluation on perplexity across different context lengths.

| Method | Proof-pile | | | | PG19 | | | |
|---|---|---|---|---|---|---|---|---|
| | 1024 | 2048 | 4096 | 8192 | 1024 | 2048 | 4096 | 8192 |
| LoRA | 3.828 | 3.369 | 3.064 | 2.867 | 9.791 | 9.098 | 8.572 | 8.199 |
| LongLoRA | 3.918 | 3.455 | 3.153 | 2.956 | 9.989 | 9.376 | 8.948 | 8.645 |
| Theta Scaling | 3.864 | 3.415 | 3.121 | 2.934 | 9.257 | 8.640 | 8.241 | 7.999 |
| TAPE | 3.641 | 3.196 | 2.901 | 2.708 | 8.226 | 7.642 | 7.278 | 7.063 |

As shown in Table 2, TAPE consistently outperforms the other methods across all context lengths on both the Proof-pile and PG19 datasets. On Proof-pile, TAPE achieves a perplexity of 3.641 at 1024 tokens, improving over LoRA (3.828), LongLoRA (3.918), and Theta Scaling (3.864). At 8192 tokens, TAPE's advantage grows, reaching 2.708, surpassing LongLoRA (2.956), LoRA (2.867), and Theta Scaling (2.934). Similarly, on PG19, TAPE achieves 8.226 at 1024 tokens, improving up to 18.3% over competitors. At 8192 tokens, TAPE reaches 7.063, further showing superiority, especially at longer context lengths.

We also evaluate the passkey retrieval accuracy of our model, following Landmark Attention (Mohtashami & Jaggi, 2023), which has also been adopted by other literature (Chen et al., 2023a;

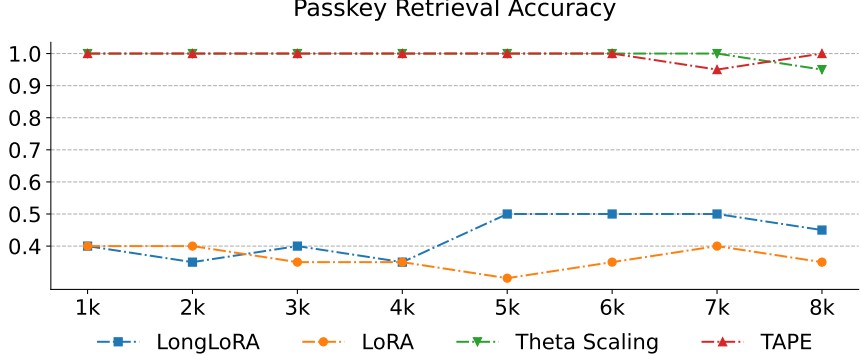

Figure 3: Accuracy on passkey retrieval from 1k to 8k context length between Llama2 7B with different fine-tuning methods.

Tworkowski et al., 2024; Chen et al., 2023b). In this task, the models are required to locate and retrieve a random passkey hidden in a long document. We test the passkey retrieval accuracy ranging from 1k to 8k. The results of long-context passkey retrieval task is presented in Figure 3. As shown, TAPE consistently achieves near-perfect accuracy across all context lengths, outperforming other methods. Theta Scaling shows a relatively stable performance while LoRA and LongLoRA exhibit fluctuating and lower accuracy. Notably, Theta Scaling is widely employed in popular open-source long-context models like Llama3 8B Instruct 262k (AI@Meta, 2024) and MistralLite (AWS, 2024). Therefore, TAPE demonstrates superior capability to be universally applied in long-context tasks.

## 4.4 EFFICIENCY ANALYSIS

In this subsection, we analyze the complexity of our methods in comparison to traditional position embedding techniques. Using the models from the pretraining experiment in Sec. 4.2, we report three key metrics: FLOPs, MACs, and the number of parameters. The metrics are evaluated with a batch size of 1 and sequence length 1024. As shown in Table 3, our architectural modifications introduce a negligible increase in FLOPs, MACs and number of parameters, compared to the standard Transformer with RoPE. Moreover, our TAPE is fully compatible with Flash Attention (Dao et al., 2022; Dao, 2024a), a widely adopted accelerated attention mechanism with IO-awareness, which introduces extra efficiency.

Table 3: Comparison of FLOPS, MACs, and parameters for models with different position embeddings.

| Method | TAPE | RoPE | FIRE | T5's relative bias |
|---|---|---|---|---|
| FLOPS (G) | 365.65 | 321.10 | 331.97 | 321.10 |
| MACs (G) | 180.69 | 160.46 | 165.69 | 160.46 |
| Params. (M) | 155.33 | 154.89 | 154.90 | 154.90 |

Table 4: System measurement. We report Execution time per step (provided in the "Time" row) and iteration per second (provided in the "throughput" row). The values are averaged over 100 inference steps.

| Method | TAPE | | RoPE | FIRE | T5's relative bias |
|---|---|---|---|---|---|
| | w/ Fusion | w/o Fusion | | | |
| Time ($\times 10^{-4}$) | 2.56 | 5.63 | 2.08 | 5.56 | 6.90 |
| Throughput | 3910 | 1775 | 4810 | 1799 | 1449 |
| Flash Attention | ✓ | ✓ | ✓ | ✗ | ✗ |

For simplicity, we evaluate the running time of attention layers with different position embedding methods on a single A100 GPU. We run 100 inference steps and report the average execution time. Both RoPE and TAPE leverage the acceleration provided by Flash Attention (Dao, 2024b), whereas FIRE and T5's relative bias are not fully compatible with Flash Attention, as it currently lacks support for gradient computation in relative bias. In contrast, we observe that the computations for position embeddings and token features in TAPE are highly parallelizable, making it suitable for further acceleration using kernel fusion techniques. To capitalize on this, we implemented a version of TAPE with kernel fusion, referred to as TAPE w/ Fusion. As shown in Table 4, the efficiency of the original TAPE (w/o Fusion) already surpasses T5's relative bias and is comparable to FIRE. With additional kernel fusion applied, TAPE achieves a $2.2\times$ speedup, approaching the efficiency of RoPE with Flash Attention.

## 5 OTHER RELATED WORK

**Length Extrapolation Technique.** The length extrapolation ability of Transformers are limited mainly in two aspects: (1) the high memory usage caused by quardratic memory usage; and (2) the poor generalizability to unseen sequence length during inference. To address the memory usage during long sequences training, LongLoRA (Chen et al., 2023b) introduced shifted sparse attention and leveraged parameter-efficient tuning. LoCoCo (Cai et al., 2024) introduce a KV cache compression mechanism. To help generalizability of positional embedding to unseen sequence length, (Chen et al., 2023a) explores zero-shot linear interpolation on rotary embedding; (r/LocalLLaMA, 2023; Peng et al., 2023) enhance simple interpolation by retaining high-frequency encoding ability; (Liu et al., 2023) investigate the relationship between rotary base and extrapolation ability. While the previously mentioned methods focus primarily on extending rotary positional embeddings, Li et al. (2023) introduced a functional relative position encoding framework that enhances generalization to longer contexts. However, these methods generally impose a fixed pattern on attention maps, often adopting a decaying pattern based on distance. In contrast, we propose a learnable and generic position encoding framework that primarily focus on arithmetic reasoning ability.

**Equivariant Machine Learning.** Developing machine learning methods that incorporate exact or approximate symmetries, such as translation and rotation, has garnered increasing interest. Convolutional neural networks, for instance, are well-known for being translation-equivariant (Sun et al., 2022), meaning that applying a translation to the input results in a corresponding transformation in the output. Broadly speaking, equivariance (with invariance as a specific case) leverages the symmetries in a problem to introduce inductive biases into neural networks, thereby reducing learning complexity and improving generalization. Prior work on equivariant machine learning has primarily focused on data with inherent symmetries, such as graphs (Wang et al., 2024a; 2022), point clouds (Zaheer et al., 2017; Qi et al., 2017), and other geometric data (Gerken et al., 2023). To the best of our knowledge, we are the first to introduce equivariance in language models, recognizing the symmetry in position embeddings.

**Generalized Rotary Embedding.** While RoPE has become widely adopted in language modeling, its potential in broader tasks remains underexplored. LieRE (Ostmeier et al., 2024) extends RoPE to 2D and 3D modalities, generalizing positional embeddings for higher-dimensional inputs. Our TAPE, when initialized as RoPE, further enhances its ability to learn adaptive positional information, focusing on text-based tasks, including more complex and position-critical challenges like arithmetic. As these works are concurrent, we believe that applying TAPE to multi-modal tasks represents a promising direction for future research.

## 6 CONCLUSION

In this paper, we introduce TAPE, a framework that enhances transformer models by contextualizing positional embeddings with sequence content across layers. Through the incorporation of permutation and orthogonal equivariance, we ensured stability and adaptability in positional encoding updates. TAPE can also be easily integrated into existing models, and introduce negligible computation and inference overhead. Extensive experiments confirmed TAPE's superiority in both arithmetic reasoning and long context language modeling task.

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

## A SETTINGS

**Hyperparameters in TAPE**  In all experiments, we set $M = 12$ and $B = 64$, with their product defining the hidden size as 768, consistent with previous work (Li et al., 2023; He et al., 2024). For TAPE, we set $L = D = 2$, consistent with the initialization of RoPE (Su et al., 2024). Additionally, we set $B' = 48$.

**Training Recipe.**  Following Brown et al. (2020), we use the causal LM objective to pretrain decoder-only Transformers with different position encodings. Our training recipe in three expriments are presented in Table 5.

Table 5: Training recipe for language model pre-training and fine-tuning in experiments.

|  | 4.1 Arithmetic | 4.2 C4 Pre-training | 4.2 SCROLLS | 4.3 Context Extension |
|---|---|---|---|---|
| Sequence length | 40 + 40 | 1024 | 1024 | 8096 |
| Batch size | 512 | 512 | 64 | 64 |
| Number of iterations | 20k | 10k | 1k | 1k |
| Attention dropout prob. | 0.0 | 0.0 | 0.0 | 0.0 |
| Optimizer | AdamW | AdamW | AdamW | AdamW |
| Learning rate | $1 \times 10^{-4}$ | $1 \times 10^{-4}$ | $1 \times 10^{-5}$ | $2 \times 10^{-5}$ |

## B ADDITIONAL EXPERIMENTS

**Ablation Study on Architecture.**  We ablate our architecture design for both attention layer and MLP layer in position contextualization. We conduct ablation studies on our architectural design for both the attention layer and the MLP layer in position contextualization. Additionally, we ablate the design of rotation equivariance by setting $\boldsymbol{W}_1 \in \mathbb{R}^{B' \times (L \cdot D)}, \boldsymbol{W}_2 \in \mathbb{R}^{(L \cdot D) \times B'}$, which disrupts the $O(D)$-equivariance, and the use of tensorial embeddings by flattening $L = D = 2$ into $L = 1$ and $D = 4$. We use the same pre-training setting as Sec. 4.2 and directly report its perplexity in test dataset of Github following He et al. (2024).

Table 6: Ablation study on TAPE architecture. We evelute pre-trained models' perplexity across varying sequence lengths on the GitHub test set.

| Architecture | | Perplexity | | | |
|---|---|---|---|---|---|
| **Attention** | **Feed Forward** | **128** | **256** | **512** | **1024** |
| ✗ | ✗ | 139.2 | 92.8 | 69.3 | 57.2 |
| ✗ | ✓ | 143.3 | 95.0 | 70.7 | 58.4 |
| ✓ | ✗ | 142.7 | 94.3 | 70.1 | 57.6 |
| ✓ | ✓ | 132.0 | 86.6 | 63.9 | 52.2 |
| **Rotation Equivariance** | **Tensorial Embedding** | | | | |
| ✓ | ✗ | 138.4 | 91.3 | 67.8 | 55.7 |
| ✗ | ✓ | 132.9 | 87.8 | 65.4 | 54.1 |
| ✓ | ✓ | 132.0 | 86.6 | 63.9 | 52.2 |

As shown in Table 6 , incorporating position contextualization in both the attention layer and the MLP layer results in the lowest perplexity across different positions within the training sequence length. Removing position contextualization from either layer increases perplexity, even exceeding that of the traditional positional embedding without any architectural modifications. This outcome is reasonable, as applying position contextualization to only one component introduces an architectural inconsistency. Furthermore, ablating rotation equivariance allows all neurons in the positional embedding to undergo linear transformations, increasing the number of parameters but leading to worse results compared to TAPE. Similarly, reducing the tensorial embedding to a vector embedding leads to higher perplexities and a decline in performance.

**Ablation Study on TAPE Hyperparameter.**   We aim to investigate the impact of varying $B'$ on learning performance. Using the same pre-training settings as described in Section 4.2, we directly report the perplexity on the GitHub test dataset. As shown in Table 7, there is no significant difference when using different values of $B'$, although a trend of first decreasing and then increasing can be observed. This suggests that a range of $B'$ values from $2B = 24$ to $3B = 48$ may yield better performance compared to other settings. Therefore, as a general guideline, we recommend considering $B' \in \{2, 3, 4\}B$ to optimize TAPE's performance.

Table 7: Ablation study on TAPE hyperparameter $B'$. We evelute pre-trained models' perplexity across varying sequence lengths on the GitHub test set.

| TAPE | | Perplexity | | | |
|---|---|---|---|---|---|
| **Added Params. (M)** | **B$'$** | **128** | **256** | **512** | **1024** |
| 0.11 | 12 | 133.2 | 87.9 | 65.2 | 53.6 |
| 0.22 | 24 | 133.0 | 86.1 | 63.2 | 51.8 |
| 0.44 | 48 | 132.0 | 86.6 | 63.9 | 52.2 |
| 0.88 | 96 | 133.2 | 87.5 | 64.5 | 52.7 |
| 1.76 | 192 | 133.0 | 87.3 | 64.5 | 53.0 |

**Stability of TAPE under Positional Shifts.**   Stability in this context refers to the consistency of a sequence's representation under positional shifts (Sun et al., 2022). To evaluate the stability of TAPE, we examine two types of positional shifts: (1) appending a [BOS] token at the beginning of the sequence and (2) initializing positional indices with non-zero values to simulate a positional translation. We analyze two aspects of the representation: the attention weights and the dot product of positional embeddings, quantifying their changes after applying positional shifts. For comparison, we include RoPE, which also exhibits $O(D)$-equivariance ($D = 2$) and remains consistent across layers, as well as TAPE without equivariance, as explored in previous ablations.

As shown in Table 8, TAPE demonstrates stability comparable to RoPE, maintaining consistent attention weights and positional embedding dot products across different layers, even under positional shifts. However, when equivariance is removed from TAPE, the differences increase significantly, especially in deeper layers, highlighting the importance of equivariance in preserving stability.

Table 8: Comparison of RoPE, TAPE, and TAPE without equivariance (W/o EQ) under positional shifts. The table shows differences in attention weights (top) and positional embedding dot products (bottom) across layers for two shift methods: adding three [BOS] tokens ("Add Tokens") and starting position IDs at 3 ("Shift IDs").

| Atten. Diff. ($\times 10^{-2}$) | Add Tokens | | | | Shift IDs | | | |
|---|---|---|---|---|---|---|---|---|
| | Layer 1 | Layer 2 | Layer 4 | Layer 8 | Layer 1 | Layer 2 | Layer 4 | Layer 8 |
| RoPE | 8.93 | 8.51 | 12.29 | 11.46 | 0.01 | 0.02 | 0.02 | 0.03 |
| TAPE | 9.08 | 11.24 | 12.23 | 13.78 | 0.01 | 0.02 | 0.04 | 0.04 |
| w/o EQ | 11.30 | 11.38 | 13.32 | 14.55 | 0.01 | 0.24 | 0.37 | 0.51 |

| PE Dot Prod. Diff. (%) | Add Tokens | | | | Shift IDs | | | |
|---|---|---|---|---|---|---|---|---|
| | Layer 1 | Layer 2 | Layer 4 | Layer 8 | Layer 1 | Layer 2 | Layer 4 | Layer 8 |
| RoPE | 0.03 | 0.03 | 0.03 | 0.03 | 0.03 | 0.03 | 0.03 | 0.03 |
| TAPE | 0.03 | 0.37 | 2.75 | 6.62 | 0.03 | 0.02 | 0.03 | 0.04 |
| w/o EQ | 0.03 | 2.29 | 3.34 | 6.37 | 0.03 | 0.54 | 0.44 | 0.86 |

**Additional Evaluation on Fine-tuned Llama-7b.**   Modern benchmarks provide a comprehensive means to assess large language models' advanced capabilities in language understanding and reasoning. Accordingly, we further evaluate our fine-tuned Llama-7b (Sec. 4.3) on standard benchmarks, including ARC (Clark et al., 2018) and MMLU (Hendrycks et al., 2021).

Table 9: Accuracy in Percentage Across Methods and Benchmarks

| Method | MMLU (%) | | | | ARC (%) | |
|---|---|---|---|---|---|---|
| | **Humanities** | **Social Sciences** | **STEM** | **Other** | **Challenge** | **Easy** |
| LoRA | $39.09 \pm 0.69$ | $46.47 \pm 0.88$ | $33.65 \pm 0.83$ | $45.83 \pm 0.89$ | $45.31 \pm 1.45$ | $74.28 \pm 0.90$ |
| LongLoRA | $37.53 \pm 0.69$ | $43.55 \pm 0.88$ | $32.54 \pm 0.83$ | $43.84 \pm 0.88$ | $45.31 \pm 1.45$ | $74.16 \pm 0.90$ |
| ThetaScaling | $37.45 \pm 0.69$ | $43.16 \pm 0.88$ | $33.05 \pm 0.83$ | $44.64 \pm 0.88$ | $45.65 \pm 1.46$ | $74.24 \pm 0.90$ |
| TAPE | $37.96 \pm 0.69$ | $45.40 \pm 0.88$ | $33.27 \pm 0.83$ | $45.06 \pm 0.88$ | $46.25 \pm 1.46$ | $74.16 \pm 0.90$ |

As Table 9 shows, TAPE demonstrates notable performance compared to other methods on MMLU and ARC benchmarks. While TAPE's accuracy on MMLU is slightly lower than that of LoRA, it consistently outperforms LongLoRA and ThetaLoRA, highlighting its strength in reasoning and language understanding. On the ARC benchmark, TAPE performs comparably to other methods on the "Easy" subset but exhibits a significant advantage on the "Challenge" subset, further underscoring its potential in complex reasoning tasks. Remarkably, these results are achieved using only fine-tuning, without pretraining TAPE, despite the presence of a certain degree of architectural shift.

**Additional Evaluation in Arithmetic Learning** We also evaluate the effectiveness of TAPE in Sec. 4.1 using a different training and testing length: 20/40 instead of 40/80. This setup is easier for the model to learn, with convergence achieved in less than half the steps. As shown in Figure 4, TAPE outperforms FIRE with a marginal improvement of 5%. However, this improvement is less pronounced compared to the case with a train/test length of 40/80, suggesting that TAPE may be more effective in tackling complex and challenging tasks than simpler ones.

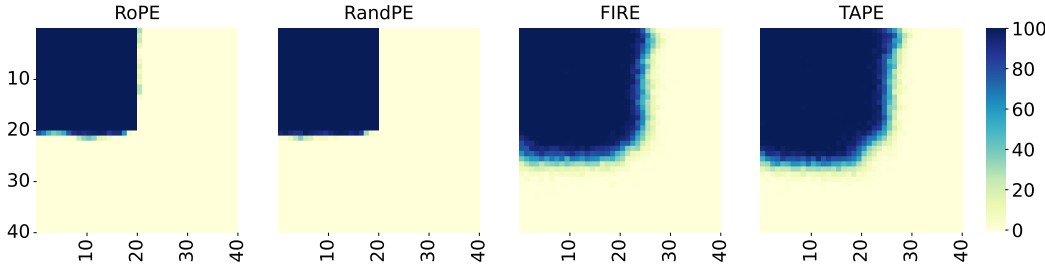

Figure 4: Accuracy on addition task trained with length 20 test on 2× context length. The average accuracy across the heatmap is 26.12%, 26.12%, 39.44% and 41.42% respectively for RoPE, RandPE, FIRE and TAPE.

**Integration with Extrapolation Technique.** Inspired by the demonstrated potential of NTK-based methods (Peng et al., 2023) to enhance the length extrapolation ability of RoPE, we have explored integrating TAPE with such techniques when initialized as RoPE. Specifically, we selected the most recent method, YaRN (Peng et al., 2023), and implemented its integration with TAPE to evaluate its performance in length extrapolation. The experiments were conducted under the same settings as described in Sec. 4.1.

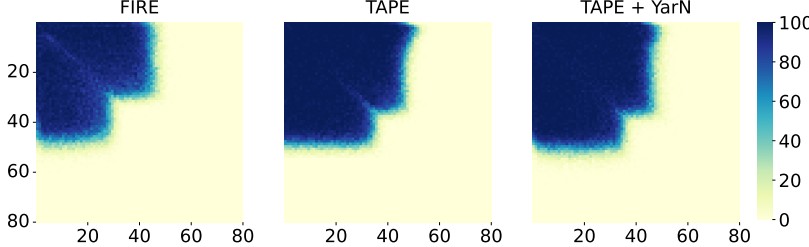

Figure 5: Accuracy on addition task between different methods on 2× context length. The average accuracy across the heatmap is 26.98%, 32.82% and 33.92% respectively for FIRE, TAPE and TAPE + YaRN.

As shown in Figure 5, the diagonal region exhibits darker colors, indicating higher accuracies. Quantitatively, YaRN effectively enhances the length extrapolation performance of TAPE with RoPE initialization, achieving a modest relative improvement of 3.4%. However, it still struggles to generalize to unseen sequences with significantly longer digit lengths.

## C  FURTHER ILLUSTRATIONS

**Illustration of Tensor Operations.**   To provide a clearer understanding of TAPE and the operation within the attention and feed-forward layers, we visualize the process in Figure 6.

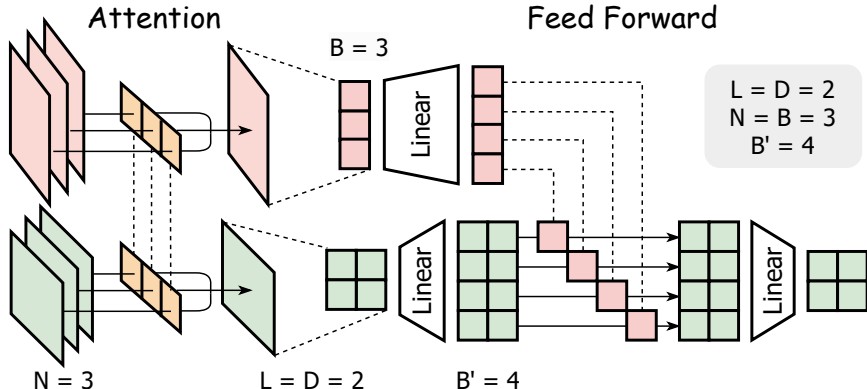

Figure 6: Illustration of TAPE's operations. The channel dimension is omitted for simplicity as all operations are channel-wise. In the attention layer, the input token embeddings have a shape of $N \times B$, and the position embeddings have a shape of $N \times L \times D$. For the feed-forward layer, the $N$ dimension is omitted as its operations are position-wise. The input token embeddings then have a shape of $B$ (or $B \times 1$), and the position embeddings have a shape of $L \times D$.

**Visualization of Attention Patterns.**   To gain insights into the effect of our proposed TAPE, we visualize the attention patterns in the last layer . We compare the attention patterns of TAPE and RoPE (Su et al., 2024). As shown in Figure 7, TAPE effectively attends to more contextual information over longer distances. In contrast, RoPE predominantly focuses on the current position, with an average attention score of $0.30$ on the diagonal of the attention patterns, compared to TAPE's $0.17$.

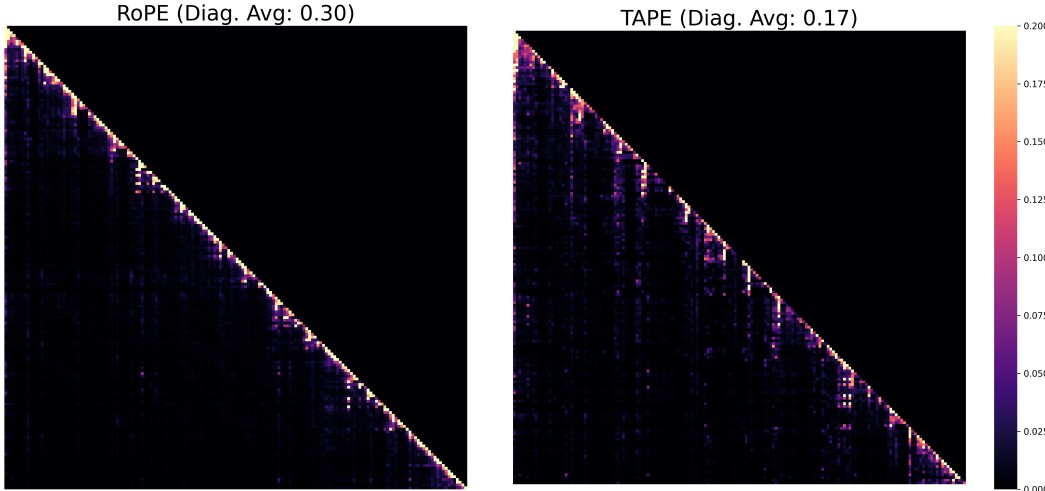

Figure 7: Comparing TAPE's attention pattern with RoPE. The sample is randomly selected from the test set of C4, with a sequence length of less than 100.

**Examples on QuALITY.** To further validate TAPE's superior performance on the SCROLLS benchmark, we present two example questions from the QuALITY dataset within the SCROLLS benchmark. As shown in Table 10 and the detailed questions in Table 11, TAPE consistently generates either the correct answer or a response similar to the correct answer, even if not an exact match. In contrast, xPos and RandPE produce meaningful sentences that are unrelated to the specific question. RoPE and ALiBi, however, generate incoherent outputs: RoPE tends to repeat certain phrases, while ALiBi fails to recognize the presence of a question, producing the same irrelevant answer regardless of the input.

Table 10: Comparing answers of different methods on example questions in QuALITY.

| Method | Question A | | Question B | |
|---|---|---|---|---|
| | Answer | EM | Answer | EM |
| Ground Truth | The secret service budget was small | ✓ | Only the private quarters or the office restroom | ✓ |
| TAPE | The secret service budget was small | ✓ | Only the private quarters | ✗ |
| xPos | They were all they were waiting for | ✗ | Only a tiny part of the right of the right to leave foreverish | ✗ |
| RandPE | Their human opinion was trusted by others who have trust the services of their people | ✗ | Only a handsome man | ✗ |
| RoPE | Their orless them together with their repories did not only they didn's never done was never done was never done... (repeating) | ✗ | The/O only the full-College All of the full-College All of the full-College... (repeating) | ✗ |
| ALiBi | Jimmy Carter is the president's de facto president | ✗ | Jimmy Carter is the president's de facto president | ✗ |

Table 11: Example Questions in QuALITY

| Qu. A (ID: 20007_RZDMZJYW_2) | Qu. B (ID: 20007_RZDMZJYW_4) |
|---|---|
| What made it easier for previous presidents to get away with adultery? | Where in the White House is it feasible for the president to meet a woman? |
| (A) Their staff did not know
(B) They always tried to hide it well
(C) The secret service budget was small
(D) The reporters never found out | (A) Only the East Wing
(B) Only the private quarters
(C) Only the oval office, bowling alley, or East Wing
(D) Only the private quarters or the office restroom |

**Article Content:**

The logistics of presidential adultery.

The Washington Times could hardly contain its excitement: "A former FBI agent assigned to the White House describes in a new book how President Clinton slips past his Secret Service detail in the dead of night, hides under a blanket in the back of a dark-colored sedan, and trysts with a woman, possibly a celebrity, at the JW Marriott Hotel in downtown Washington." For Clinton-haters, Gary Aldrich's tale sounded too good to be true. And it was.

The not-so-Secret-Service agent's "source" turned out to be a thirdhand rumor passed on by Clinton scandal-monger David Brock. Those who know about White House security—Clinton staffers, the Secret Service, former aides to Presidents Reagan and Bush—demolished Aldrich's claims. Clinton couldn't give his Secret Service agents the slip (they shadow him when he walks around the White House), couldn't arrange a private visit without tipping off hotel staff, and couldn't re-enter the White House without getting nabbed. (Guards check all cars at the gate—especially those that arrive at 4 a.m.)

Even so, the image resonates. For some Americans, it is an article of faith: Bill Clinton cheated on his wife when he was governor, and he cheats on her as president. But can he? Is it possible for the president of the United States to commit adultery and get away with it? Maybe, but it's tougher than you think.

Historically, presidential adultery is common. Warren Harding cavorted with Nan Britton and Carrie Phillips. Franklin Roosevelt "entertained" Lucy Rutherford at the White House when Eleanor was away. America was none the wiser, even if White House reporters were.

Those who know Clinton is cheating often point to the model of John F. Kennedy, who turned presidential hanky-panky into a science. Kennedy invited mistresses to the White House for afternoon (and evening, and overnight) liaisons. Kennedy seduced women on the White House staff (including, it seems, Jackie's own press

*Continued on next page...*

secretary). Kennedy made assignations outside the White House, then escaped his Secret Service detail by scaling walls and ducking out back doors. If Kennedy did it, so can Clinton.

Well, no. Though Clinton slavishly emulates JFK in every other way, he'd be a fool to steal Kennedy's MO d'amour. Here's why:

1) Too many people would know. Kennedy hardly bothered to hide his conquests. According to Kennedy mistress (and mob moll) Judith Campbell's autobiography, those who knew about their affair included: Kennedy's personal aides and secretary (who pandered for him), White House drivers, White House gate guards, White House Secret Service agents, White House domestic staff, most of Campbell's friends, a lot of Kennedy's friends, and several Kennedy family members. Such broad circulation would be disastrous today because:

2) The press would report it. Kennedy conducted his affairs brazenly because he trusted reporters not to write about them. White House journalists knew about, or at least strongly suspected, Kennedy's infidelity, but never published a story about it. Ask Gary Hart if reporters would exercise the same restraint today. Clinton must worry about this more than most presidents. Not only are newspapers and magazines willing to publish an adultery story about him, but many are pursuing it.

For the same reason, Clinton would find it difficult to hire a mistress. A lovely young secretary would set off alarm bells in any reporter investigating presidential misbehavior. Says a former Clinton aide, "There has been a real tendency to have no good-looking women on the staff in order to protect him."

3) Clinton cannot avoid Secret Service protection. During the Kennedy era, the Secret Service employed fewer than 500 people and had an annual budget of about $4 million. Then came Lee Harvey Oswald, Squeaky Fromme, and John Hinckley. Now the Secret Service payroll tops 4,500 (most of them agents), and the annual budget exceeds $500 million (up 300 percent just since 1980). At any given time, more than 100 agents guard the president in the White House. Top aides from recent administrations are adamant: The Secret Service never lets the president escape its protection.

So what's a randy president to do? Any modern presidential affair would need to meet stringent demands. Only a tiny number of trusted aides and Secret Service agents could know of it. They would need to maintain complete silence about it. And no reporters could catch wind of it. Such an affair is improbable, but—take heart, Clinton-haters—it's not impossible. Based on scuttlebutt and speculation from insiders at the Clinton, Bush, Reagan, and Ford White Houses, here are the four likeliest scenarios for presidential adultery. 1) The White House Sneak. This is a discreet variation of the old Kennedy/Campbell liaison. It's late at night. The president's personal aides have gone home. The family is away. He is alone in the private quarters. The private quarters, a.k.a. "the residence," occupy the second and third floors of the White House. Secret Service agents guard the residence's entrances on the first floor and ground floors, but the first family has privacy in the quarters themselves. Maids and butlers serve the family there, but the president and first lady ask them to leave when they want to be alone. The president dials a "friend" on his private line. (Most presidents placed all their calls through the White House operators, who kept a record of each one; the Clintons installed a direct-dial line in the private quarters.) The president invites the friend over for a cozy evening at the White House. After he hangs up with the friend, he phones the guard at the East Executive Avenue gate and tells him to admit a visitor. He also notifies the Secret Service agent and the usher on duty downstairs that they should send her up to the residence. A taxi drops the woman near the East gate. She identifies herself to the guard, who examines her ID, runs her name through a computer (to check for outstanding warrants), and logs her in a database. A White House usher escorts her into the East Wing of the White House. They walk through the East Wing and pass the Secret Service guard post by the White House movie theater. The agent on duty waves them on. The usher takes her to the private elevator, where another Secret Service agent is posted. She takes the elevator to the second floor. The president opens the door and welcomes her. Under no circumstances could she enter the living quarters without first encountering Secret Service agents.

Let us pause for a moment to demolish two of the splashier rumors about White House fornication. First, the residence is the only place in the White House where the president can have safe (i.e., uninterrupted) sex. He can be intruded upon or observed everywhere else—except, perhaps, the Oval Office bathroom. Unless the president is an exhibitionist or a lunatic, liaisons in the Oval Office, bowling alley, or East Wing are unimaginable. Second, the much-touted tunnel between the White House and the Treasury Department is all-but-useless to the presidential adulterer. It is too well-guarded. The president could smuggle a mistress through it, but it would attract far more attention from White House staff than a straightforward gate entry would.

Meanwhile, back in the private quarters, the president and friend get comfortable in one of the 14 bedrooms (or, perhaps, the billiard room). After a pleasant 15 minutes (or two hours?), she says goodbye. Depending on how long she stays, she may pass a different shift of Secret Service agents as she departs. She exits the White House grounds, unescorted and unbothered, at the East gate.

The Risks: A gate guard, an usher, and a handful of Secret Service agents see her. All of them have a very good idea of why she was there. The White House maid who changes the sheets sees other suspicious evidence. And the woman's—real—name is entered in a Secret Service computer. None of this endangers the president too much. The computer record of her visit is private, at least for several decades after he leaves office. No personal aides know about the visit. Unless they were staking out the East gate, no journalists do either. The Secret Service agents, the guard, the steward, and the maid owe their jobs to their discretion. Leaks get them fired.

That said, the current president has every reason not to trust his Secret Service detail. No one seriously compares Secret Service agents (who are pros) to Arkansas state troopers (who aren't). But Clinton might not trust any

*Continued on next page...*

security guards after the beating he took from his Arkansas posse. Also, if other Secret Service agents are anything like Aldrich, they may dislike this president. One Secret Service leak—the lamp-throwing story—already damaged Clinton. Agents could tattle again.

2) The "Off-the-Record" Visit. Late at night, after his personal aides and the press have gone home, the president tells his Secret Service detail that he needs to take an "off-the-record" trip. He wants to leave the White House without his motorcade and without informing the press. He requests two agents and an unobtrusive sedan. The Secret Service shift leader grumbles but accepts the conditions. Theoretically, the president could refuse all Secret Service protection, but it would be far more trouble than it's worth. He would have to inform the head of the Secret Service and the secretary of the Treasury.

The president and the two agents drive the unmarked car to a woman friend's house. Ideally, she has a covered garage. (An apartment building or a hotel would raise considerably the risk of getting caught.) The agents guard the outside of the house while the president and his friend do their thing. Then the agents chauffeur the president back to the White House, re-entering through the Southwest or Southeast gate, away from the press station.

The Risks: Only two Secret Service agents and their immediate supervisor know about the visit. It is recorded in the Secret Service log, which is not made public during the administration's tenure. Gate guards may suspect something fishy when they see the car. A reporter or passer-by could spy the president—even through tinted windows—as the car enters and exits the White House. The friend's neighbors might spot him, or they might notice the agents lurking outside her house. A neighbor might call the police to report the suspicious visitors. All in all, a risky, though not unthinkable, venture.

3) The Camp David Assignation. A bucolic, safer version of the White House Sneak. The president invites a group of friends and staffers—including his paramour but not his wife—to spend the weekend at Camp David. The girlfriend is assigned the cabin next to the president's lodge. Late at night, after the Hearts game has ended and everyone has retired to their cabins, she strolls next door. There is a Secret Service command post outside the cabin. The agents on duty (probably three of them) let her enter. A few hours later, she slips back to her own cabin.

The Risks: Only a few Secret Service agents know about the liaison. Even though the guest list is not public, all the Navy and Marine personnel at Camp David, as well as the other guests, would know that the presidential entourage included an attractive woman, but not the first lady. That would raise eyebrows if it got back to the White House press room.

4) The Hotel Shuffle. The cleverest strategy, and the only one that cuts out the Secret Service. The president is traveling without his family. The Secret Service secures an entire hotel floor, reserving elevators and guarding the entrance to the president's suite. The president's personal aide (a man in his late 20s) takes the room adjoining the president's. An internal door connects the two rooms, so the aide can enter the president's room without alerting the agents in the hall. This is standard practice. Late in the evening, the aide escorts a comely young woman back to the hotel. The

Secret Service checks her, then waves her into the aide's room. She emerges three hours later, slightly disheveled. She kisses the aide in the hall as she leaves. Someone got lucky—but who?

The Risks: The posted Secret Service agents might see through the charade. More awkwardly, the aide would be forced to play the seamy role of procurer. (He would probably do it. Kennedy's assistants performed this task dutifully.)

In short, presidential adultery is just barely possible in 1996. But it would be extremely inconvenient, extremely risky, and potentially disastrous. It seems, in fact, a lot more trouble than it's worth. A president these days might be wiser to imitate Jimmy Carter, not Jack Kennedy, and only lust in his heart.

