# OpenReview forum: "Rethinking Addressing in Language Models via Contextualized Equivariant Positional Encoding"
_ICLR.cc/2025/Conference — Submitted to ICLR 2025_

### Official Review · Reviewer_NG6y · 2024-10-18

**Soundness:** 3
**Presentation:** 3
**Contribution:** 4
**Rating:** 8
**Confidence:** 4

**Summary:**

The paper introduces TAPE, a novel positional encoding that dynamically adapts to sequence content. By incorporating context-aware encodings across layers, TAPE more effectively captures long-range dependencies. Experiments show its superior performance on long-text tasks.

**Strengths:**

1. The paper introduces TAPE, a remarkable method that uses contextualized information to update positional encodings.
2. TAPE is a plug-and-play approach that can be seamlessly integrated into common LLMs based on RoPE.
3. Experimental results show that TAPE significantly enhances long-context processing capabilities.
4. The paper is well-written and easy to follow.

Overall, I think this is a valuable work. TAPE effectively integrates positional and semantic information through multiplicative positional encoding to improve model performance. Unlike traditional static encodings or those modified solely by attention scores, this method allows positional encodings to evolve dynamically with layer depth, similar to hidden states.

**Weaknesses:**

The authors did not remove the empty Appendix section in the template.

**Questions:**

1. Could the authors provide additional results on length extrapolation (training on short sequences, testing on longer ones) and short-text capabilities, such as MMLU?

2. If an LLM is trained with TAPE, is directly extending the model's context window as challenging as with RoPE? If extending the context window is difficult, are there methods similar to NTK or PI to achieve effective and efficient context window expansion?

3. Could the authors conduct ablation studies on the impact of the Attention and MLP blocks in updating TAPE?

---

> ### Author Response · Authors · 2024-11-22
> **Response to Reviewer NG6y**
>
> We greatly thank Reviewer NG6y for appreciating our contributions, providing valuable suggestions on improving the work, and supporting the acceptance of this work. We address the questions as follows.
>
> > Q1: Could the authors provide additional results on length extrapolation (training on short sequences, testing on longer ones) and short-text capabilities, such as MMLU?
>
> Thank you for bringing this up. We have included evaluation results, including MMLU, in Table 7, Appendix B. The results demonstrate that TAPE performs competitively on these tasks, even when the base model is not pre-trained with the same architecture. For details on length extrapolation, please refer to our response to Q2.
>
> > Q2: If an LLM is trained with TAPE, is directly extending the model's context window as challenging as with RoPE? If extending the context window is difficult, are there methods similar to NTK or PI to achieve effective and efficient context window expansion?
>
> Thank you for raising this question. In Section 4.1, we evaluate TAPE’s zero-shot length extrapolation ability using an arithmetic learning task, where the test length is twice the training length. TAPE demonstrates competitive performance in this setup. Inspired by your excellent suggestion, we have explored YarN [1] to further enhance TAPE’s length extrapolation capability in Appendix B. As shown in Figure 5, YaRN effectively enhances the length extrapolation performance of TAPE with RoPE initialization, achieving a relative improvement of 3.4%.
>
> > Q3: Could the authors conduct ablation studies on the impact of the Attention and MLP blocks in updating TAPE?
>
> Yes, we have included ablation studies in the Appendix B to validate our design choices for incorporating position contextualization in both the attention and MLP layers. We evaluate models’ perplexity results on the test set use the same pretraining setup described in Sec. 4.2. A summarized version of Table 6 in Appendix B is provided below, showing that removing any component increases perplexity.
> | Attention | MLP | Perplexity  |
> | --- | --- | --- |
> | ✗ | ✓ | 58.4 |
> | ✓ | ✗ | 57.6 |
> | ✓ | ✓ | 52.2 |
>
> **References**:
>
> [1] Peng, Bowen, et al. "YaRN: Efficient Context Window Extension of Large Language Models." The Twelfth International Conference on Learning Representations.

---

> > ### Comment · Reviewer_NG6y · 2024-11-22
> > **Keeping my scores.**
> >
> > Thank you for your rebuttal, I will keep my scores.

---

### Official Review · Reviewer_8dnP · 2024-10-24

**Soundness:** 2
**Presentation:** 3
**Contribution:** 2
**Rating:** 3
**Confidence:** 4

**Summary:**

This paper proposes a new method - TAPE - to incorporate context into positional encodings to enhance model's ability to reason over longer context. Authors propose to 1) extend positional encodings to a multi-dimensional format, by dividing each token into M segments, and assigning  each segment a matrix-form positional embedding. Authors show that RoPE is a special case of TAPE; 2) update positional encodings using attention layers' weights. Experiments on one Arithmetic, four long-context and two language modeling tasks demonstrate that TAPE can outperform the baselines used in the paper.

**Strengths:**

1. The proposed method of incorporating into positional encodings is novel.
2. TAPE method does not require significant additional computations and can be incorporated into existing models during fine-tuning.
3. Proposed approach shows improvements across several evaluation tasks.

**Weaknesses:**

Overall, the motivation and design principles are clear. However, the experimental part lacks thorough evaluation, ablation, and explanation, in particular:
1. In Section 3.2 Authors introduce a number of hyperparameters, specific to TAPE. However in 4.1 and 4.2, where models are trained from scratch, authors do not specify how TAPE was initialized. For ex, what are M, B, L, D, B', E, etc? How those where selected and why? In finetuning things are more clear, where TAPE was initialized to align with RoPE , but what's the size of W1/2 (B`) and how it was selected?
3. Authors perform evaluation on selected tasks claiming "superior performance in language modeling, arithmetic reasoning, and long-context retrieval tasks":
a. Arithmetic learning task: superior performance belongs to Abacus Embeddings (McLeish et al., 2024), that can solve problem with up to 100 operands. Regarding embeddings including in the comparison, I find information in caption and in text confusing: on pictures, FIRE and TAPE look almost identical, your report "The average accuracy across the heatmap is 26.12%, 26.12%, 49.44% and 41.42% respectively for RoPE, RandPE, FIRE and TAPE." - i.e. TAPE performs *worse* than FIRE. But then in the text of Section 4.1 authors claim "Compared to FIRE, which achieves 25.24% ..., TAPE shows a remarkable 51.9% relative improvement." - which seem to contradict the picture.
b. Evaluation on SCROLLS task is further confusing: this benchmark contains 6 datasets, with test partitions kept in private. Authors report performance only on 4 selected datasets out of 6. Not clear what data where actually used for evaluation, and what was the evaluation setup. It's even more surprising to see low scores on QuALITY benchmark, which is multi-choice QA dataset, where random guess achieves 25% accuracy, while the best performance reported is 11.6%. If authors used another evaluation pipeline, it should be thoroughly described in the paper.
c. Finetuning: authors report perplexity on language modeling task, and accuracy on Passkey retrieval, but did not investigate performance on standard reasoning evaluation benchmarks, or even long-context QA reported in section 4.2.
With all that, author's claim about "TAPE’s superiority in both arithmetic reasoning and long context language modeling task" is questionable.
4. Performance capability beyond 155M to be explored.
5. Authors did not perform or report any ablations on hyperparameters introduced in the paper, or how those were selected.

It worth noting that tensorial positional encodings idea seems to be very similar to LieRE embeddings, that also introduces multi-dimensional rotation matrix  (Ostmeier, 2024).

Finally, there are multiple typos across section 4 that need to be fixed before publishing (see "Questions" section).



References:
1. Ostmeier, Sophie, et al. "LieRE: Generalizing Rotary Position Encodings." arXiv preprint arXiv:2406.10322 (2024)
2. Sean McLeish, Arpit Bansal, Alex Stein, Neel Jain, John Kirchenbauer, Brian R. Bartoldson, Bhavya
Kailkhura, Abhinav Bhatele, Jonas Geiping, Avi Schwarzschild, and Tom Goldstein. Transformers can do arithmetic with the right embeddings. arXiv preprint arXiv:2405.17399, (2024)

**Questions:**

1. Why do you think TAPE outperforms other baselines on long-context tasks, especially on QuALITY?

+ See "Weaknesses" section.

Some typos:
Line 311: " inEq. 8" -> " in Eq. 8" (space)
Lines 326-328: "rely absolute" -> rely on absolute;  (Sec. 3.3). ->  (Sec. 4.3).
Line 359: " demonstrate TAPE ’s superior" ->  demonstrate TAPE’s superior (space)
Line 376: "RoPE (Kitaev et al., 2020) ALiBi" -> "RoPE (Kitaev et al., 2020), ALiBi" (comma)
Line 451: "of parameters The" -> "of parameters. The"
Line 479: "Inn con" -> "In con"
Line 483: "Table 4 , the" -> "Table 4, the"
Line 503: " we proposes" ->  we propose

---

> ### Author Response · Authors · 2024-11-22
> **Response to Reviewer 8dnP (1/2)**
>
> We greatly thank Reviewer 8dnP for reviewing our work and provide insightful suggestions on further strengthening the manuscript. We address the concerns as follows.
>
> > W1: In Section 3.2 Authors introduce a number of hyperparameters, specific to TAPE. However in 4.1 and 4.2, where models are trained from scratch, authors do not specify how TAPE was initialized. For ex, what are M, B, L, D, B', E, etc? How those where selected and why? In finetuning things are more clear, where TAPE was initialized to align with RoPE , but what's the size of W1/2 (B`) and how it was selected?
>
> Thank you for pointing this out. The values M = 12, B = 64, B’ = 48, and L = D = 2 are used consistently across all experiments, as clarified in the updated Appendix. Specifically, L = D = 2 was chosen to align with RoPE, while M = 12 and B = 64 are consistent with prior work. B’ = 48 selected to introduce a minor (~1%) increase in parameters.
>
> > W2: Authors perform evaluation on selected tasks claiming "superior performance in language modeling, arithmetic reasoning, and long-context retrieval tasks":
> > 1. Arithmetic learning task: superior performance belongs to Abacus Embeddings (McLeish et al., 2024), that can solve problem with up to 100 operands. Regarding embeddings including in the comparison, I find information in caption and in text confusing: on pictures, FIRE and TAPE look almost identical, your report "The average accuracy across the heatmap is 26.12%, 26.12%, 49.44% and 41.42% respectively for RoPE, RandPE, FIRE and TAPE." - i.e. TAPE performs worse than FIRE. But then in the text of Section 4.1 authors claim "Compared to FIRE, which achieves 25.24% ..., TAPE shows a remarkable 51.9% relative improvement." - which seem to contradict the picture.
> > 2. Evaluation on SCROLLS task is further confusing: this benchmark contains 6 datasets, with test partitions kept in private. Authors report performance only on 4 selected datasets out of 6. Not clear what data where actually used for evaluation, and what was the evaluation setup. It's even more surprising to see low scores on QuALITY benchmark, which is multi-choice QA dataset, where random guess achieves 25% accuracy, while the best performance reported is 11.6%. If authors used another evaluation pipeline, it should be thoroughly described in the paper.
> > 3. Fine-tuning: authors report perplexity on language modeling task, and accuracy on Passkey retrieval, but did not investigate performance on standard reasoning evaluation benchmarks, or even long-context QA reported in section 4.2. With all that, author's claim about "TAPE’s superiority in both arithmetic reasoning and long context language modeling task" is questionable.
>
> We sincerely appreciate your detailed feedback and aim to address the points raised as follows:
> 1. **Arithmetic Learning**: Sorry for the confusion caused by the caption mismatch. We have corrected the error and aligned the caption description with the main text. For a fair comparison, we do not consider abacus embeddings, although they can be seamlessly integrated with TAPE. Additionally, we included an extra baseline for this experiment, and our method still achieves SOTA performance in the revised manuscript.
> 2. **Comprehensive Evaluation on SCROLLS**:  The SCROLLS benchmark includes three additional summarization tasks (GovReport, QMSum, and SummScreenFD), which were initially omitted due to the higher inference time required compared to QA tasks. To provide comprehensive results, we have now completed evaluations across all 7 tasks and included the results in Table 1. A summarized version of the results below shows that TAPE achieves the best overall performance. Regarding the evaluation setup, we used the official evaluation code on the validation dataset, which will be detailed in our open-sourced code. Regarding QuALITY, please refer to response to Q1 for a clarification on its validity.
> | Method | QAS | CNLI | NQA | QuAL | QMS | SumS | GovR |
> | --- | --- | --- | --- | --- | --- | --- | --- |
> | RoPE | 8.39 | 65.00 | 1.77 | 0.04 | 6.34 | 5.63 | 9.71 |
> | ALiBi | 8.25 | 69.62 | 4.11 | 0.0 | 9.92 | 9.78 | 18.81 |
> | RandPE | 13.44 | 62.01 | 4.63 | 0.38 | 8.43 | 8.31 | 8.93 |
> | xPos | 9.02 | 71.75 | 4.83 | 0.24 | 10.73 | 9.38 | 16.38 |
> | TAPE (ours) | 11.52 | 72.80 | 6.79 | 11.60 | 12.42 | 10.34 | 15.18 |
> 3. **More Evaluations on Fine-tuned Models**: We have added evaluations on standard reasoning and language understanding benchmarks, ARC and MMLU, in Table 7, Appendix B. The results demonstrate that TAPE performs competitively on these tasks, even when the base model is not pre-trained with the same architecture. Regarding SCROLLS, which requires a fine-tuned model for evaluation [1,2,3], we are currently conducting these resource-intensive experiments and plan to include the results in the camera-ready version, time permitting.

---

> > ### Author Response · Authors · 2024-11-22
> > **Response to Reviewer 8dnP (2/2)**
> >
> > > W3: Performance capability beyond 155M to be explored.
> >
> > Thank you for raising this point. We are currently replicating the experiment in Sec. 4.2 on a larger model (~400M) following FIRE [3]. Since training a single method (with Flash Attention) on each dataset requires over 500 GPU-hours, we will update the results here as soon as we obtain reliable comparisons and promise to include the full results in the camera-ready version. In theory, TAPE is expected to maintain its superiority, given its significant, not minor, performance boost. Moreover, performance rankings across different scales tend to remain consistent, as evidenced by the rank consistency of about 0.9 for seven methods in FIRE [3]. Additionally, we highlight that recent works on positional embeddings [1,3,4,5] have not been tested on models exceeding 500M parameters. Notably, we are the first to present results on a 7B LLaMA model, utilizing adaptation techniques enabled by the TAPE framework.
> >
> > > W4: Authors did not perform or report any ablations on hyperparameters introduced in the paper, or how those were selected.
> >
> > Please refer to our response to W1.
> >
> >
> > > Q1: Why do you think TAPE outperforms other baselines on long-context tasks, especially on QuALITY?
> >
> > We would like to clarify the validity of our results on the QuALITY dataset. QuALITY is a multiple-choice dataset, but in SCROLLS [2], it is evaluated as a natural language generation task using the Exact Match metric, as implemented in its official code repository ([link](https://huggingface.co/datasets/tau/scrolls/blob/main/metrics/scrolls.py#L113)). The challenge lies in requiring the model to generate text that exactly matches the correct choice, rather than selecting from predefined options. The weak performance of models on QuALITY aligns with previous work [1,3]. For example, as shown in Table 1 of BiPE [1], the low Exact Match rates (<1%) for RoPE and ALiBi are consistent with our reported results.
> >
> > > It worth noting that tensorial positional encodings idea seems to be very similar to LieRE embeddings, that also introduces multi-dimensional rotation matrix (Ostmeier, 2024).
> >
> > Thank you for highlighting this work. LieRE extends RoPE to 2D and 3D modalities, focusing on generalizing positional embeddings for higher-dimensional inputs. In contrast, our TAPE enhances the expressive power of positional embeddings, enabling Transformers to address adaptive positional information, with empirical evidence demonstrating its effectiveness in various text tasks. As these works are concurrent, we believe applying TAPE to multi-modal tasks is a promising direction for future research. These discussions have also been added to the related work section.
> >
> > > Finally, there are multiple typos across section 4 that need to be fixed before publishing.
> >
> > We have corrected all the typos mentioned above and reviewed the revised manuscript. Thank you for your attention to detail!
> >
> > **References**:
> >
> > [1] He, Zhenyu, et al. "Two Stones Hit One Bird: Bilevel Positional Encoding for Better Length Extrapolation." Forty-first International Conference on Machine Learning. 2024.
> >
> > [2] Shaham, Uri, et al. "SCROLLS: Standardized CompaRison Over Long Language Sequences." Conference on Empirical Methods in Natural Language Processing. 2022.
> >
> > [3] Li, Shanda, et al. "Functional Interpolation for Relative Positions improves Long Context Transformers." The Twelfth International Conference on Learning Representations. 2023.
> >
> > [4] Ostmeier, Sophie, et al. "LieRE: Generalizing Rotary Position Encodings." arXiv preprint arXiv:2406.10322 (2024).
> > [5] McLeish, Sean, et al. "Transformers Can Do Arithmetic with the Right Embeddings." arXiv preprint arXiv:2405.17399 (2024).

---

> > ### Comment · Reviewer_8dnP · 2024-11-22
> > **Thank you for your responses**
> >
> > I would like to thank authors for their updates and responses.
> > I'm not sure my concerns were fully addressed though:
> >
> > W1 >> M = 12 and B = 64 are consistent with prior work
> >
> > Can you please specify which prior work? In the updated appendix you refer to (Li et al, 2023, Chen at al, 2023b) - neither of these works used mentioned parameters, did they? While either of these parameters can take values between 1 and 758 constrained their product stayed the same, there are a lot of options to consider. And B' adds another degree of freedom to learn. How did you choose these parameters? Have you done any ablations?
> >
> > W2 1 >> We have corrected the error and aligned the caption description
> >
> > I'm comparing the previous version of Fig 2, and current, and they look quite different. Both FIRE and TAPE accuracy patterns and numbers are different from what was reported in the original version, and FIRE results are different from what was reported previously (McLeish et al., 2024). Can you please clarify where all these changes and differences are coming from?
> >
> > W2 2 >> please refer to response to Q1
> >
> > Looking at the reference [1] you provided in you response, training setup seems to be the same. From page 6 in [1] "Settings": "We choose the standard decoderonly Transformer as the base model .. . For the Transformer-based language model, we
> > set the number of layers to 12, the hidden dimension to 768, and the number of attention heads to 12. The total number of model parameters is approximately 155M." - which is exactly the same setup you are describing on page 8, lines 390-394. However, results reported on baselines in Tables 1 in both papers are quite different. Are there some hyperparameters missing that bring this difference? For example there is a 10x difference in RoPE's and 30x in RandPE accuracy on QuAL. Since you have referenced that paper, it would be nice to include BiPE-RoPEyarn results as well for fair comparison.
> >
> > W2 3 >> in Table 7, Appendix B.
> >
> > Do I read it correctly, that improvements in perplexity (Table 2) didn't result in improvements in accuracy? Isn't the later more important? If so, worth moving to the main text.
> > >> Regarding SCROLLS, which requires a fine-tuned model for evaluation [1,2,3] - not sure I understand, why would it require finetuning? You have reported pre-training evaluations in Table 1
> >
> > W4 >> Please refer to our response to W1.
> >
> > I don't think my question was answered. There are no ablations.
> >
> > Q1: I don't think my question was answered. Results on SCROLLS are very positive, while not so much in finetuning on other benchmarks. Why do you think TAPE outperforms other baselines on long-context tasks, especially on QuALITY?

---

> > > ### Author Response · Authors · 2024-11-24
> > > **Further Response to Reviewer 8dnP (1/3)**
> > >
> > > Thank you for your questions. We hope the following answers address your concerns, and we are willing to provide further clarifications if necessary.
> > >
> > > > W1: Can you please specify which prior work? In the updated appendix you refer to (Li et al, 2023, Chen at al, 2023b) - neither of these works used mentioned parameters, did they? While either of these parameters can take values between 1 and 758 constrained their product stayed the same, there are a lot of options to consider. And B' adds another degree of freedom to learn. How did you choose these parameters? Have you done any ablations?
> > > >
> > >
> > > The prior works referenced are BiPE [1] and FIRE [3]. For clarity, we have included the relevant tables from these works as below. Specifically, the "attention heads" parameter corresponds to M, and the "head dimensions" parameter corresponds to B.
> > >
> > > Regarding the ablations for B′, we conducted experiments with values of 12, 24, 48, 96, and 192, as shown in Table 7. For convenience, we have summarized these results in the following table. The findings indicate minimal performance variation across different values of B′. However, configurations where B’=2B=24 or B’=4B=48 appear to yield slightly better results.
> > >
> > > ### Ablations in B’
> > >
> > > | B' | PPL (128) | PPL (1024) |
> > > | --- | --- | --- |
> > > | 12 | 133.2 | 53.6 |
> > > | 24 | 133.0 | 51.8 |
> > > | 48 | 132.0 | 52.2 |
> > > | 96 | 133.2 | 52.7 |
> > > | 192 | 133.0 | 53.0 |
> > >
> > > ### Table 8 in FIRE [3]
> > > |  | Small model |
> > > | --- | --- |
> > > | Training sequence length | 2048 |
> > > | Number of layers | 12 |
> > > | Attention heads | 12 |
> > > | Hidden layer size | 768 |
> > > | Head dimensions | 64 |
> > > | FFN activation | GeLU |
> > > | Number of parameters | 125M |
> > >
> > > ### Table 5 in BiPE [1]
> > > ---
> > > |  |  |
> > > | --- | --- |
> > > | Layers | 12 |
> > > | Attention heads | 12 |
> > > | Head dimensions | 64 |
> > > | Hidden dimensions | 768 |
> > > | FFN dimensions | 3072 |
> > > | Model parameters | 155M |
> > >
> > > > W2.1: I’m comparing the previous version of Fig 2, and current, and they look quite different. Both FIRE and TAPE accuracy patterns and numbers are different from what was reported in the original version, and FIRE results are different from what was reported previously (McLeish et al., 2024). Can you please clarify where all these changes and differences are coming from?
> > > >
> > >
> > > The differences arise from adjustments to the training settings. Specifically, we increased the task difficulty by changing the training/test sequence lengths from 20/40 to 40/80. This modification makes the tasks more challenging for the model to learn and requires more steps to converge. We made this change because TAPE shows greater advantages in handling more complex and difficult tasks, as also indicated by the ARC-challenge results in Table 7. Thank you for pointing out the omission of the original results. For completeness, we have included the original results under the "easy" setting in Appendix B.
> > >
> > > > W2.2: Looking at the reference [1] you provided in you response, training setup seems to be the same. From page 6 in [1] "Settings": "We choose the standard decoderonly Transformer as the base model .. . For the Transformer-based language model, we set the number of layers to 12, the hidden dimension to 768, and the number of attention heads to 12. The total number of model parameters is approximately 155M." - which is exactly the same setup you are describing on page 8, lines 390-394. However, results reported on baselines in Tables 1 in both papers are quite different. Are there some hyperparameters missing that bring this difference? For example there is a 10x difference in RoPE's and 30x in RandPE accuracy on QuAL. Since you have referenced that paper, it would be nice to include BiPE-RoPEyarn results as well for fair comparison.
> > > >
> > >
> > > Our setup is not identical to [1] as our training recipe differs, which is detailed in Table 5 of our revised version compared to Table 6 in BiPE [1]. Specifically, we use a shorter training schedule with 10k steps, a batch size of 0.5M tokens, and a total of 5B training tokens. This adjustment approximately aligns with the scaling laws [4] (see Table 3 in Chinchilla [4] and Table 12 in Mamba [5]) and makes the experiments more computationally affordable. For clarity, we have included these tables as below for reference.
> > >
> > > Regarding BiPE as a baseline, it is important to note that BiPE introduces a general framework that can theoretically be applied to any method. Therefore, we view it more as a complementary approach than a direct competitor.
> > >
> > > ### Table 12 in Chinchilla [4]: Scaling Law Model Sizes.
> > > | Params | Training steps | Batch Size | Tokens |
> > > | --- | --- | --- | --- |
> > > | 125M | 4800 | 0.5M tokens | 2.5B |
> > > | 350M | 13500 | 0.5M tokens | 7B |
> > > | 760M | 29000 | 0.5M tokens | 15B |
> > > | 1.3B | 50000 | 0.5M tokens | 26B |
> > >
> > > ---
> > >
> > > ### Table 3 in Mamba [5]: Estimated optimal training tokens for various model sizes.
> > > | Parameters | Tokens |
> > > | --- | --- |
> > > | 400 Million | 8.0 Billion |
> > > | 1 Billion | 20.2 Billion |
> > > | 10 Billion | 205.1 Billion |
> > > | 67 Billion | 1.5 Trillion |

---

> ### Author Response · Authors · 2024-11-24
> **Further Response to Reviewer 8dnP (2/3)**
>
> > W2.3 >> in Table 7, Appendix B. Do I read it correctly, that improvements in perplexity (Table 2) didn't result in improvements in accuracy? Isn't the later more important? If so, worth moving to the main text.
> >
>
> You are correct that improvements in perplexity (Table 2) do not necessarily translate to improvements in accuracy on standard benchmarks. This is because accuracy in such benchmarks primarily depends on the pre-trained base model and can be further enhanced by techniques like RLHF. The aim of the experiment in Table 2 is not to improve benchmark accuracy directly but to evaluate which techniques enable a model pre-trained on short-length text to adapt effectively to long-length text through fine-tuning, without the need for pre-training on long text from scratch. Perplexity is used here as a metric to assess the model's adaptation to long text, as demonstrated in prior work like LongLoRA [6].
>
> > W4: I don't think my question was answered. There are no ablations.
> >
>
> Please refer to our response to W1 for the specification of other parameters and ablations for B′.
>
> > Regarding SCROLLS, which requires a fine-tuned model for evaluation [1,2,3] - not sure I understand, why would it require finetuning? You have reported pre-training evaluations in Table 1
> >
>
> We would like to confirm that SCROLLS does require fine-tuning. Pre-training a language model on a corpus and then fine-tuning it on SCROLLS is the standard approach in previous works [1, 2, 3], as the benchmark is specifically designed for fine-tuning.
>
> Our experiments in Section 4.2 follow a similar setup: we pre-train the model from scratch and then fine-tune and test it on downstream SCROLLS. To avoid any confusion, we’ve added further clarification in the revised paper.
>
> For additional context, we also explored how zero-shot generation looks for Llama-7b fine-tuned with LoRA. The outputs in QuALITY were largely nonsensical, with repeated patterns like “The\n\u0409\n\u0409\n\u0409...” until the token limit was reached. This suggests that the QuALITY benchmark is not suited for zero-shot or few-shot generation and instead requires fine-tuning for meaningful evaluation.
>
> > Q1: Why do you think TAPE outperforms other baselines on long-context tasks, especially on QuALITY?
> >
>
> To understand why TAPE outperforms other baselines on long-context tasks like QuALITY, it is important to consider the challenges faced by Transformers with existing positional embeddings in such scenarios. Traditional positional embeddings often assume a distance-decay effect, where information from words farther apart is given less significance. While this assumption works well for most language tasks, it may fail in long-context scenarios where critical information could appear at the beginning or at positions far from the output.
>
> TAPE addresses this limitation by incorporating positional contextualization, allowing positional embeddings to adaptively adjust the importance of different positions without relying on a fixed decay pattern. This is supported by our visualization of attention patterns in Figure 7, which shows that TAPE can effectively attend to longer contexts.
>
> For further illustration, we compared TAPE’s output with other baselines on two specific questions from QuALITY. As shown in the following table, TAPE consistently generates either the correct answer or a response close to the correct one, unlike other methods. Related analyses and the specific questions are detailed in Table 10 and Table 11 of Appendix C.
>
> | **Method** | **Question A** - Answer | **EM** | **Question B** - Answer | **EM** |
> | --- | --- | --- | --- | --- |
> | Ground Truth | The secret service budget was small | ✔ | Only the private quarters or the office restroom | ✔ |
> | TAPE | The secret service budget was small | ✔ | Only the private quarters | ✗ |
> | xPos | They were all they were waiting for | ✗ | Only a tiny part of the right of the right to leave foreverish | ✗ |
> | RandPE | Their human opinion was trusted by others who have trust the services of their people | ✗ | Only a handsome man | ✗ |
> | RoPE | Their orless them together with their repories did not only they didn’s never done was never done was never done... (repeating) | ✗ | The/O only the full-College All of the full-College All of the full-College... (repeating) | ✗ |
> | ALiBi | Jimmy Carter is the president’s de facto president | ✗ | Jimmy Carter is the president’s de facto president | ✗ |

---

> > ### Author Response · Authors · 2024-11-24
> > **Further Response to Reviewer 8dnP (3/3)**
> >
> > **References**:
> >
> > [1] He, Zhenyu, et al. "Two Stones Hit One Bird: Bilevel Positional Encoding for Better Length Extrapolation." Forty-first International Conference on Machine Learning. 2024.
> >
> > [2] Shaham, Uri, et al. "SCROLLS: Standardized CompaRison Over Long Language Sequences." Conference on Empirical Methods in Natural Language Processing. 2022.
> >
> > [3] Li, Shanda, et al. "Functional Interpolation for Relative Positions improves Long Context Transformers." The Twelfth International Conference on Learning Representations. 2023.
> >
> > [4] Hoffmann, Jordan, et al. "Training compute-optimal large language models." arXiv preprint arXiv:2203.15556 (2022).
> >
> > [5] Gu, Albert, and Tri Dao. "Mamba: Linear-time sequence modeling with selective state spaces." First Conference on Language Modeling. 2024.
> >
> > [6] Chen, Yukang, et al. "LongLoRA: Efficient Fine-tuning of Long-Context Large Language Models." The Twelfth International Conference on Learning Representations. 2023.

---

> ### Author Response · Authors · 2024-11-30
>
> Dear Reviewer 8dnP,
>
> We want to thank you for the constructive comments in your review. As a follow-up on our responses, we would like to kindly remind you that the discussion period is ending soon. We hope to use this open response period to discuss the paper to solve the concerns and improve the quality of our paper. Have you gotten a chance to read our further responses and revision, which attempt to address all of your concerns? In the revised manuscript, we have included additional experiments and illustrations, such as hyperparameter ablation studies, and example QA from QuALITY. For your convenience, we have also provided detailed references in our response. We sincerely hope to have further discussions with you to see if our response solves the concerns. We would be more than happy to provide more information or clarification, should it be necessary. We hope our paper, as the first work to propose the integration of position contextualization into an enhanced Transformer, could be valued and receive a positive and fair assessment.
>
> Best,
>
> Paper 1199 Authors

---

> > ### Author Response · Authors · 2024-12-02
> >
> > Dear Reviewer 8dnP,
> >
> > We apologize for following up again so soon, but with the discussion period ending shortly, we wanted to kindly check if you’ve had a chance to check our responses and revisions which try to address your concerns.
> >
> > We greatly appreciate your time and effort, and we’re happy to provide any additional clarification if needed. Thank you again for your understanding and thoughtful review.
> >
> > Best,
> >
> > Paper 1199 Authors

---

### Official Review · Reviewer_nfEP · 2024-11-03

**Soundness:** 3
**Presentation:** 3
**Contribution:** 3
**Rating:** 8
**Confidence:** 5

**Summary:**

This paper introduces a new positional embedding method called TAPE, which extends the RoPE positional embedding to be both learnable and context-dependent. Experimental results show that TAPE achieves better performance. Additionally, the authors developed a kernel to reduce TAPE's runtime latency, which could be highly beneficial if released as open-source.

**Strengths:**

This paper systematically generalizes the RoPE positional embedding. The authors begin by introducing two essential properties that a positional embedding design should satisfy, then propose TAPE to meet these criteria. I especially appreciate the kernel implementation, which significantly reduces TAPE's runtime. I've seen several complex positional embedding designs, and a common drawback is their slow runtime in practice. I'm glad the authors of TAPE have addressed this issue.

**Weaknesses:**

1. **Illustration of TAPE Method**: It would be highly beneficial if the authors could include an illustration of the TAPE method. Visualizing it with a block diagonal matrix and context-dependent data flow should be feasible and would greatly enhance readers' understanding of TAPE’s operations.

2. **Missing Hyperparameters**: Some important hyperparameters are not specified. For instance, is \( B = L = D = 2 \) used across all experiments? Additionally, details on hyperparameters like learning rate, batch size, and optimizer are missing and should be included.

3. **Potential Error in Figure 2 Caption**: The caption for Figure 2 may contain an error. It states, "The average accuracy across the heatmap is 26.12%, 26.12%, 49.44%, and 41.42% respectively for RoPE, RandPE, FIRE, and TAPE." This appears to contradict line 364, which mentions that "TAPE shows a remarkable 51.9% relative improvement."

4. **Explanation for Improved Arithmetic Task Performance**: More explanation is needed on why TAPE improves performance on arithmetic tasks. Are the authors suggesting that TAPE has inherent advantages in this area?

5. **Clarification on TAPE’s Task Performance**: It’s unclear *how* TAPE improves performance across all reported tasks. Does TAPE enable more flexible attention patterns? Could you, for instance, visualize the \(\phi\) function block by block in equation (6) to provide further insights?

6. **Compatibility of T5 with Flash Attention**: I’m unsure if it's accurate to state that T5 is incompatible with flash attention. While support is currently lacking, compatibility could be achieved with implementation. For instance, see [here](https://github.com/Dao-AILab/flash-attention/pull/956) and [here](https://github.com/Dao-AILab/flash-attention/issues/332).

**Questions:**

See above. In addition, will the code be released?

---

> ### Author Response · Authors · 2024-11-22
> **Response to Reviewer nfEP**
>
> We greatly thank Reviewer nfEP for appreciating our contributions, providing valuable suggestions on improving the readability, and supporting the acceptance of this work. We address the concerns as follows.
>
> > W1: Illustration of TAPE Method: It would be highly beneficial if the authors could include an illustration of the TAPE method. Visualizing it with a block diagonal matrix and context-dependent data flow should be feasible and would greatly enhance readers' understanding of TAPE’s operations.
>
> Thank you for this excellent suggestion. We have added an illustration figure (Figure 6) in Appendix C to better visualize TAPE’s operations, including the context-dependent data flow in both attention layer and MLP layer. Please feel free to share any further ideas for improving the visualization.
>
> > W2: Missing Hyperparameters: Some important hyperparameters are not specified. For instance, is ( B = L = D = 2 ) used across all experiments? Additionally, details on hyperparameters like learning rate, batch size, and optimizer are missing and should be included.
>
> Thank you for pointing this out. L = D = 2 is used across all experiments, as clarified in the updated Appendix A. Additional training hyperparameters, including learning rate, batch size, and optimizer details, have also been provided.
>
> > W3: Potential Error in Figure 2 Caption: The caption for Figure 2 may contain an error. It states, "The average accuracy across the heatmap is 26.12%, 26.12%, 49.44%, and 41.42% respectively for RoPE, RandPE, FIRE, and TAPE." This appears to contradict line 364, which mentions that "TAPE shows a remarkable 51.9% relative improvement."
>
> Sorry for the confusion caused by the caption mismatch. We have corrected the error and aligned the caption description with the main text. Additionally, we included an extra baseline for this experiment, and our method still achieves SOTA performance in the revised manuscript.
>
> > W4: Explanation for Improved Arithmetic Task Performance: More explanation is needed on why TAPE improves performance on arithmetic tasks. Are the authors suggesting that TAPE has inherent advantages in this area?
>
> Thank you for bringing this up. Here is a more detailed explanation:
>
> In arithmetic tasks, every digit has equal importance to the equation, regardless of its distance from the output. Traditional positional embeddings often assume a distance-decay effect, where words farther apart are less significant in the output. While this assumption is valid for most language tasks, it does not hold for arithmetic tasks. Positional contextualization enables dynamic reweighting of positional importance based on the task context, preserving effective distance decay for language tasks while addressing arithmetic contexts appropriately. This highlights TAPE’s potential advantages in arithmetic tasks.
>
> We have also included this explanation in Sec. 4.1 to further clarify TAPE’s advantages in arithmetic tasks.
>
> > W5: Clarification on TAPE’s Task Performance: It’s unclear how TAPE improves performance across all reported tasks. Does TAPE enable more flexible attention patterns? Could you, for instance, visualize the (\phi) function block by block in equation (6) to provide further insights?
>
> Yes, thank you for raising this excellent point. We have added a visualization of the last layer’s attention patterns in Figure 7, Appendix C, showing that TAPE effectively captures and attends to more contextual information over longer distances compared to RoPE.
>
> > W6: Compatibility of T5 with Flash Attention: I’m unsure if it's accurate to state that T5 is incompatible with flash attention. While support is currently lacking, compatibility could be achieved with implementation.
>
> Thank you for pointing this out. We agree that T5 could theoretically be made compatible with Flash Attention, but it has not yet been implemented. We have softened our tone in the main text to reflect this. To clarify, the main engineering challenge lies in the difficulty of implementing backward computation with gradient calculations for relative position embeddings, which is why Alibi is currently the only supported relative position embedding. In contrast, our TAPE is natively compatible with Flash Attention and could achieve even greater efficiency through our customized fused kernel, which we consider a notable practical contribution.
>
> > Will the code be released?
>
> Yes, we will open-source the code immediately and make the model available in the near future.

---

> ### Comment · Reviewer_nfEP · 2024-11-22
> **Thank you for the response.**
>
> Dear authors,
>
> Thank you for your response. I increased my score to 8 and lean toward accepting this paper.

---

### Official Review · Reviewer_SBfN · 2024-11-04

**Soundness:** 2
**Presentation:** 2
**Contribution:** 3
**Rating:** 5
**Confidence:** 4

**Summary:**

Increased the score from 3 -> 5 after rebuttal.

====

In the current Transformer architecture, the interaction between positional embedding (PE) and token embedding (TE) is simplistic and favors methods that mainly rely on tokens (e.g. natural language generation) rather than tasks that require positions (e.g. addition). The paper proposes better integration of PE and TE. The proposed method TAPE, updates PE along with TE throughout the layers such that the functions that contextualize PE and TE with respect to each other obey constraints such as permuation invariance and orthogonal equivariance. The main motivation for the method is to improve the stability of PEs during updates. The method shows competitive performance on tasks that require position indexing.

**Strengths:**

1. The problems with current PE and the related work on PE are explained nicely.
2. The idea of imposing constraints such that updates to PE and TE are permutation invariant and orthogonal equivariant is neat.

**Weaknesses:**

1. The complexity of the method is not justified. For example, it is not clear why there should be PEs corresponding to blocks of input? And why should the PE be a tensor?
2. The method is motivated in terms of increasing stability of PEs, but there is no empirical evidence presented. The paper lacks many ablations that justify the design decisions of the new architecture.
3. The method increases the number of parameters. A fair comparison should be provided for baselines by making sure the number of parameters is same.
4. It is unclear if the performance of the proposed method is stronger than existing proposals. For example, based on Figure 2 (caption), it seems to me that the method performs poorly than FIRE.
5. Baselines such as NoPE (no position encoding) and FIRE are missing for experiments (NoPE for Sec 4.1 and 4.2, and FIRE for Sec 4.2).
6. The method is mainly motivated for permutaion invariance but all evaluation is performed with causal models which are permutation variant.

**Questions:**

1. Is the description of Figure 2 incorrect because it doesn't match the description provided in lines 361--365.
2. Where are the ablation results for design decisions? For example, what happens if you treat PE as a vector instead of tensor, what if the PE is the same for all blocks?
3. Can you visualize attention patterns or some qualitative analysis on where the proposed method shines?
4. The paper mainly motivates that TAPE ensures "stability." What do you mean by that and how do you measure it? Where is the empirical evidence for that?

---

> ### Author Response · Authors · 2024-11-22
> **Response to Reviewer SBfN (1/2)**
>
> We greatly thank Reviewer SBfN for reviewing our work and providing insightful suggestions on further strengthening the manuscript. We address the concerns as follows.
> > W1: The complexity of the method is not justified. For example, it is not clear why there should be PEs corresponding to blocks of input? And why should the PE be a tensor?
>
> Thank you for pointing this out. Here, we will justify our motivation for this design as follows. Ablation results validating the effectiveness of updating PEs and using tensorial PEs can be found in our response to Q2.
> - To clarify our motivation for incorporating positional contextualization, it is important to understand the limitations of Transformers in addressing arithmetic tasks. Traditional positional embeddings often assume a distance-decay effect, where words farther apart have less significance in the output. While this assumption holds for most language tasks, it does not apply to arithmetic tasks, where every digit has equal importance regardless of its distance. Positional contextualization allows us to dynamically reweight positional importance based on the task context, maintaining effective distance decay for language tasks while handling arithmetic contexts appropriately.
> - Regarding the design of tensorial embeddings versus vector embeddings, a simple way to understand their relationship is to view a tensor as a multi-channel vector. A tensor with a single channel reduces to a vector. This technique of increasing dimensionality to enhance expressive power has been adopted in several works, including RoPE [1], which extends sinusoidal PE to a 2-channel format (one for sine and the other for cosine), and the multi-head attention mechanism [2] in the Transformer.
>
> > W2: The method is motivated in terms of increasing stability of PEs, but there is no empirical evidence presented. The paper lacks many ablations that justify the design decisions of the new architecture.
>
> We have included ablation studies in the Appendix B to validate our design choices for the attention and MLP layers, as well as for the tensorial PEs and equivariant PEs to ensure stability. For a detailed analysis of PE stability, please refer to our response to Q4.
>
> > W3: The method increases the number of parameters. A fair comparison should be provided for baselines by making sure the number of parameters is same.
>
> While our method does increase the number of parameters, the difference is very minor—only about 1.1%—as shown in Table 3 of Sec. 4.4 (155.3M compared to 154.9M). In prior research on large language models [1,2], models with parameter counts of 125M, 130M, and 160M are often considered to be of equivalent scale. Moreover, this minor parameter increase leads to significant performance gains, such as a 22% relative improvement over the previous SOTA method in the arithmetic learning task (Sec. 4.1).
>
> > W4: It is unclear if the performance of the proposed method is stronger than existing proposals. For example, based on Figure 2 (caption), it seems to me that the method performs poorly than FIRE.
>
> Sorry for the confusion caused by the caption mismatch. Our method actually achieves SOTA performance in this experiment. We have fixed the caption description and added an extra baseline for this experiment in the revised manuscript.
>
> > W5: Baselines such as NoPE (no position encoding) and FIRE are missing for experiments (NoPE for Sec 4.1 and 4.2, and FIRE for Sec 4.2).
>
> Thanks for bringing this up. We have included NoPE in Sec 4.1 and FIRE in Sec 4.2, and our method continues to demonstrate the best overall performance in these experiments. We will also complete running NoPE results and add them to Sec 4.2 in the camera-ready version if time permits.
>
> > W6: The method is mainly motivated for permutaion invariance but all evaluation is performed with causal models which are permutation variant.
>
> Thank you for pointing this out. While our experiments focus on causal language tasks, it is important to note that Transformers are inherently permutation-invariant architectures. They are commonly applied to causal tasks using causal masks and objectives during pre-training. Since our method aims to improve the positional addressing capabilities of Transformers, ensuring compatibility with their intrinsic permutation invariance is essential. We selected language tasks for evaluation because of their wide applicability and impact, even though order-irrelevant tasks could also be explored in future work.

---

> ### Author Response · Authors · 2024-11-22
> **Response to Reviewer SBfN (2/2)**
>
> > Q1: Is the description of Figure 2 incorrect because it doesn't match the description provided in lines 361--365.
>
> Yes, this has been addressed as noted in W4.
>
> > Q2: Where are the ablation results for design decisions? For example, what happens if you treat PE as a vector instead of tensor, what if the PE is the same for all blocks?
>
> We have included thorough ablation results for design decisions Appendix B. These results cover scenarios such as using vector PE and keeping the same PE across all blocks, i.e., not incorporating positional contextualization in all layers. The models’ perplexity on the test set was evaluated using the same pretraining setup described in Sec. 4.2. A summarized version of Table 6 in Appendix B is shown below, demonstrating that any ablation increases perplexity.
>
> | Updating PEs | Tensorial PEs | Perplexity |
> | --- | --- | --- |
> | ✗ | ✓ | 57.2 |
> | ✓ | ✗ | 55.7 |
> | ✓ | ✓ | 52.2 |
>
> > Q3: Can you visualize attention patterns or some qualitative analysis on where the proposed method shines?
>
> Yes, thank you for raising this excellent point. We have added a visualization of the last layer’s attention patterns in Figure 7, Appendix C. The results show that TAPE effectively captures and attends to more contextual information over longer distances.
>
> > Q4: The paper mainly motivates that TAPE ensures "stability." What do you mean by that and how do you measure it? Where is the empirical evidence for that?
>
> By "stability," we mean that the representation of a sequence remains consistent under positional translation, as defined in prior work [3]. Specifically, this refers to maintaining the same sentence embedding even when padding is added at the beginning or end of the sequence. For instance, when a shifted sentence is input into a Transformer with rotary positional embedding, the positional embedding rotates in its embedding space. Our method ensures rotational equivariance, enabling the same output after encoding. Empirical evidence is presented in our ablation study in Appendix B, where removing rotational equivariance from layer updates results in a decline in performance (increasing perplexity) compared to TAPE. A quick overview of this result is provided in the summary table below.
>
> | Equivariance | Perplexity |
> | --- | --- |
> | ✗ | 54.1 |
> | ✓ | 52.2 |
>
> **References**:
>
> [1] Biderman, Stella, et al. "Pythia: A suite for analyzing large language models across training and scaling." International Conference on Machine Learning. PMLR, 2023.
>
> [2] Gu, Albert, and Tri Dao. "Mamba: Linear-time sequence modeling with selective state spaces." First Conference on Language Modeling. 2024.
>
> [3] Sun, Yutao, et al. "A Length-Extrapolatable Transformer." Proceedings of the 61st Annual Meeting of the Association for Computational Linguistics (Volume 1: Long Papers). 2023.

---

> ### Comment · Reviewer_SBfN · 2024-11-25
>
> Thank you for your response and revisions. Some of these details are important which were absent in the original submission.
>
> > Q4: "stability." What do you mean by that and how do you measure it? Where is the empirical evidence for that?
>
> The padding experiment in the rebuttal is useful. What is the result with no padding, i.e., do you end up with the same number with and without padding? This is important too to make your point that TAPE is orthogonal variant. I am surprised that there is no definition of stability in the original submission. Your ablation in the experiment reminded me of this paper [1].
>
> > Q2: Where are the ablation results for design decisions?
>
> In your response, why is there no row where both of them are absent? I could not find it in the revision too. Is that equivalent to ROPE? Some of this discussion has to be in the main paper, not appendix.
>
> > W6: The method is mainly motivated for permutaion invariance
>
> I don't find this response satisfying. Motivating the entire paper with permutation invariance but choosing an evaluation setting with Transformer decoder is contradicting. Is there a way you can prove this point empirically?
>
> Many of these details are not present in the original submission, and the revision provides some of these details in the appendix. I am increasing the score to 5 as I still believe the paper can be improved a lot with respect to its motivations and linking them to experimental results/ablations. I will not be able to engage further due to bandwidth.
>
> [1] The curious case of absolute position embeddings. Sinha et al. 2022

---

> ### Author Response · Authors · 2024-12-02
> **Further response to Reviewer SBfN (1/2)**
>
> Thank you for your questions. We hope the following answers address your concerns, and we are willing to provide further clarifications if necessary.
>
> > Q2: Where are the ablation results for design decisions? → In your response, why is there no row where both of them are absent? I could not find it in the revision too. Is that equivalent to ROPE? Some of this discussion has to be in the main paper, not appendix.
> >
>
> There are no such a setting that both are set as “w/o” because you can find in our ablation setting that the L = 2 in equivariance ablation while L = 1 in tensor ablation, so these two configurations are not orthogonal and not directly combined. Note that we carefully need to find a representative way to ablate each design aspect—equivariance and tensorial embedding—without introducing excessive variation, as these were originally integrated into our dedicated design. As suggested, we have found that setting L = 4 and D = 1  ablates both equivariance and tensor simultaneously. The results are provided below.
>
> | Rotation Equivariance | Tensorial PEs | Perplexity |
> | --- | --- | --- |
> | ✗ | ✗ | 56.2 |
> | ✓ | ✗ | 55.7 |
> | ✗ | ✓ | 54.1 |
> | ✓ | ✓ | 52.2 |
>
> > Q4: What do you mean by "stability" and how do you measure it? Where is the empirical evidence for that? → The padding experiment in the rebuttal is useful. What is the result with no padding, i.e., do you end up with the same number with and without padding? This is important too to make your point that TAPE is orthogonal variant. I am surprised that there is no definition of stability in the original submission. Your ablation in the experiment reminded me of this paper [1].
> >
>
> In the revised manuscript, we have added the explanation (definition) for "stability" in the main text and provided empirical evidence, including comprehensive padding results, to show TAPE’s stability in Appendix B. Thank you for pointing out this work [1]. We find it a good example for emphasizing the importance of permutation invariance in positional embeddings, as absolute position embeddings are unable to model the permutation-invariant relationships between positional information. This insight aids in understanding our response to W6.
>
> > W6: The method is mainly motivated for permutaion invariance → I don't find this response satisfying. Motivating the entire paper with permutation invariance but choosing an evaluation setting with Transformer decoder is contradicting. Is there a way you can prove this point empirically?
> >
>
> We would like to clarify that permutation invariance is not the primary motivation; our method is primarily driven by positional contextualization. Permutation invariance, in turn, serves to enhance positional contextualization by ensuring stable updates. This is because it aligns with the permutation-invariant nature of Transformers and is embedded in several previous embedding methods, making it a natural principle for us to adopt without unnecessarily disrupting it.
>
> Moreover, we would like to further explain the rationale behind prioritizing language tasks over permutation-invariant ones. While we acknowledge that permutation invariance is crucial for tasks where it is required, our model is not restricted to permutation-invariant tasks. Transformers, when combined with a simple causal mask, can handle language tasks, and our enhanced model can be applied in the same way. Therefore, experiments on a variety of tasks are useful for demonstrating the model's effectiveness, and it is not necessary to restrict them to permutation-invariant tasks. We prioritize language tasks to showcase the effectiveness of positional contextualization, as this technique is primarily motivated by the need to address complex language tasks, such as arithmetic and long-context reasoning.

---

> ### Author Response · Authors · 2024-12-02
> **Further response to Reviewer SBfN (2/2)**
>
> To provide a clearer understanding of our work, it is crucial to differentiate between two key concepts that should not be conflated as “our method”: our design principles and the enhanced Transformer. To avoid such confusion, we commit to including these clarifications in a future version of our paper.
>
> 1. Our design principles for Transformers with positional contextualization include permutation invariance. Many previous positional embedding methods, such as RoPE and other relative embeddings, also adhere to this principle. The essence of these methods is that positional relationships matter, but absolute positional order does not, meaning positional information should be permutation-invariant. We adopt this principle in our enhanced Transformer, not only because it aligns with the permutation-invariant nature of Transformers, but also because it reflects the permutation-invariant nature of positional information itself. Empirical evidence supporting the use of this principle in positional embeddings can be found in the ineffectiveness of absolute position embeddings, which are permutation-variant, as demonstrated in [1, 2].
> 2. For our enhanced Transformer, the primary motivation is positional contextualization—updating positional embeddings through the Transformer architecture. This motivation arises from the limitations of Transformers in handling complex language tasks, such as arithmetic and long-context reasoning. Accordingly, we choose to evaluate our architecture on these language tasks. While our model is general and can handle other permutation-invariant tasks more effectively, this is beyond the scope of our current motivation and remains a promising direction for future work.
>
> **References**
>
> [1] Sinha, Koustuv, et al. "The Curious Case of Absolute Position Embeddings." *Findings of the Association for Computational Linguistics: EMNLP 2022*. 2022.
>
> [2] Wang, Yu-An, and Yun-Nung Chen. "What Do Position Embeddings Learn? An Empirical Study of Pre-Trained Language Model Positional Encoding." *Proceedings of the 2020 Conference on Empirical Methods in Natural Language Processing (EMNLP)*. 2020.

---

### Author Response · Authors · 2024-11-22
**General Response**

Dear reviewers and AC:

We sincerely thank all the reviewers for their time and effort in reviewing our paper. Based on their valuable feedback, we have revised the manuscript and uploaded the updated version. Below is a summary of the changes made:

- **Expanded Experiments**:

    In response to the reviewers' suggestions for more comprehensive experiments, we have added additional baselines and datasets to enhance our evaluation:

    - In Sec. 4.1, NoPE was added as a baseline.
    - In Sec. 4.2, FIRE was added, along with three additional datasets in SCROLLS benchmark. (Note: The result for FIRE on one new dataset is still running.)
- **Additional Details and Illustrations**:

    To address the reviewers' suggestions and deepen the understanding of TAPE’s benefits, we have added three new sections in the Appendix to provide additional details and insights:

  - **Appendix A**: Outlines the experimental setup, including the training recipe and hyperparameter configurations for TAPE.
  - **Appendix B**: Presents supplementary experiments, including ablation studies validating the design of equivariant and tensorial embeddings, analyzing the effects of Attention and MLP layers, and investigating the impact of hyperparameter choices. This section also provides empirical evidence of TAPE's stability, evaluations on ARC and MMLU language benchmarks for the fine-tuned model (Sec. 4.3), and demonstrates the integration of TAPE with YaRN to improve length extrapolation.
  - **Appendix C**: Includes visualizations of TAPE operations, illustrations of attention patterns, and example question-answering results from QuALITY to further elucidate TAPE’s mechanisms and superiority.

- **Enhanced Text Clarity**:

  We are grateful for the reviewers' detailed feedback and suggestions for improving the text. We have incorporated these refinements to enhance the manuscript's readability, including clearer explanations of stability, improved discussions of TAPE's advantages in arithmetic tasks, and other related enhancements.

If there are any additional questions or further suggestions, we would be happy to address them promptly. Thank you once again for your time and effort in reviewing our work.


Best,

Paper 1199 Authors

---

### Meta-Review · Area_Chair_4x9E · 2024-12-23

**Metareview:**

### Summary
The paper proposes TAPE, for improving positional embeddings in Transformers by integrating context-aware positional encodings across layers. TAPE proposes an integration of positional embedding (PE) and token embedding (TE) in a novel way. The work is supported by  experiments in SCROLLs, parameter-efficient finetuning, etc, to demonstrate TAPE's effectiveness.

### Strengths
The proposed approach is interesting and clear, addressing limitations in traditional and existing advanced methods. Furthermore, the authors offer practical contributions, such as parameter-efficient integration and compatibility with advanced architectures like Flash Attention, and provide well-structured experiments.

### Weaknesses
However, the paper has notable weaknesses. First, as a paper proposing new positional embedding method, I think one standard and important experiment is evaluating PPL w.r.t. different context lengths for pre-training. However, authors only report fine-tuned results on SCROLLs. While authors reported PPL with parameter-efficient finetuning, it's not a standard setting for a new method in positional embedding.

In addition, the results of TAPE reported in the paper are much better than other baselines (e.g. SCROLLs and PEFT PPL), while I am not clear how the baseline results are produced. For example, the results of LongLoRA and Scaling Theta are worse than LoRA. When finetuning with LoRA, it's unclear if authors have adjusted the base hyperparameter of RoPE. I think one reliable setting might be finding some published results reported in other positional embedding papers and report results following exactly the same setting.

Reviewers also raised questions regarding lack of comprehensive ablations for some hyperparameters and the presentation of experimental results and setup was unclear. Additionally, while the authors attempt to address permutation invariance, its empirical validation remains incomplete. Some results on arithmetic and long-context tasks appear inconsistent with baselines, raising questions about reproducibility and experimental transparency.

**Additional Comments On Reviewer Discussion:**

Several main points were raised during rebuttal:

1. Experimental Setup and Results Clarity: Reviewers raised concerns about inconsistencies and insufficient details in the experimental setup, particularly some results in the SCROLLS benchmarks. The authors provided clarifications, corrected errors, and included additional results to improve transparency, however I think the concerns are not fully addressed.

2. Ablation Studies: The lack of ablations for specific hyperparameters (e.g., B′, M, L) and design decisions was a significant concern. The authors added detailed ablation studies to Appendix, showing the impact of their choices on perplexity and contextualization.

3. Permutation Invariance and Task Choice: Reviewers questioned the relevance of permutation invariance given the use of causal language tasks. While the authors clarified their motivation, empirical validation of this property was limited. They emphasized positional contextualization as the primary goal.

I think the biggest issue of this paper is unclear experimental setups of baselines while presenting seemingly much better results and lack of PPL w.r.t. context lengths experiments with pre-training.

---

### Decision · Program_Chairs · 2025-01-22

Reject